# Concentration-Dependent Global Quantitative Proteome Response of *Staphylococcus epidermidis* RP62A Biofilms to Subinhibitory Tigecycline

**DOI:** 10.3390/cells11213488

**Published:** 2022-11-03

**Authors:** Kidon Sung, Miseon Park, Jungwhan Chon, Ohgew Kweon, Saeed A. Khan, Andrew Shen, Angel Paredes

**Affiliations:** 1Division of Microbiology, National Center for Toxicological Research, US FDA, Jefferson, AR 72079, USA; 2Companion Animal Health, Inje University, Gimhae 50834, Korea; 3Division of Neurotoxicology, National Center for Toxicological Research, US FDA, Jefferson, AR 72079, USA; 4Office of Scientific Coordination, National Center for Toxicological Research, US FDA, Jefferson, AR 72079, USA

**Keywords:** global quantitative proteome, *S. epidermidis* RP62A, biofilm, tigecycline

## Abstract

*Staphylococcus epidermidis* is a leading cause of biofilm-associated infections on implanted medical devices. During the treatment of an infection, bacterial cells inside biofilms may be exposed to sublethal concentrations of the antimicrobial agents. In the present study, the effect of subinhibitory concentrations of tigecycline (TC) on biofilms formed by *S. epidermidis* strain RP62A was investigated using a quantitative global proteomic technique. Sublethal concentrations of TC [1/8 (T1) and 1/4 minimum inhibitory concentration (MIC) (T2)] promoted biofilm production in strain RP62A, but 1/2 MIC TC (T3) significantly inhibited biofilm production. Overall, 413, 429, and 518 proteins were differentially expressed in biofilms grown with 1/8 (T1), 1/4 (T2), and 1/2 (T3) MIC of TC, respectively. As the TC concentration increased, the number of induced proteins in each Cluster of Orthologous Groups (COG) and Kyoto Encyclopedia of Genes and Genomes (KEGG) pathway increased. The TC concentration dependence of the proteome response highlights the diverse mechanisms of adaptive responses in strain RP62A biofilms. In both COG and KEGG functional analyses, most upregulated proteins belong to the metabolism pathway, suggesting that it may play an important role in the defense of strain RP62A biofilm cells against TC stress. Sub-MIC TC treatment of strain RP62A biofilms led to significant changes of protein expression related to biofilm formation, antimicrobial resistance, virulence, quorum sensing, ABC transporters, protein export, purine/pyrimidine biosynthesis, ribosomes, and essential proteins. Interestingly, in addition to tetracycline resistance, proteins involved in resistance of various antibiotics, including aminoglycosides, antimicrobial peptides, β-lactams, erythromycin, fluoroquinolones, fusidic acid, glycopeptides, lipopeptides, mupirocin, rifampicin and trimethoprim were differentially expressed. Our study demonstrates that global protein expression profiling of biofilm cells to antibiotic pressure may improve our understanding of the mechanisms of antibiotic resistance in biofilms.

## 1. Introduction

*Staphylococcus epidermidis* is a coagulase-negative staphylococcus that is linked to prosthetic valve endocarditis, prosthetic joint infections, pacemaker electrode infections, and catheter-associated bloodstream infections [1,2]. It can attach to the surface of medical devices and develop biofilms, which are composed of teichoic acids, lipids, exopolysaccharides, proteins, and extracellular DNA [3]. Polysaccharide intercellular adhesin (PIA) (*icaA, icaB, icaC, icaD*), extracellular matrix binding protein (*ebh, embP*), accumulation-associated protein (*aap, sesF*), and biofilm-associated protein homolog (*bhp, sesD*) are involved in biofilm formation of *S. epidermidis* [4].

Bacteria within biofilms can be protected from host immune defenses and antibiotic therapies [5]. Compared to planktonic cells, they have been reported to be over 1000 times more resistant to antimicrobial agents [6]. They have subpopulations with different phenotypic levels of antimicrobial or stress resistance, termed persister cells, and slow, nongrowing or viable but nonculturable cells with different metabolic activities because of non-uniform gradients of pH, oxygen, water and nutrients inside the biofilm matrix [7]. In addition, extracellular polymeric substances of biofilms can prevent the penetration of antibiotics into the biofilm interior, causing bacterial cells in this location to be exposed to sublethal concentrations of the antibiotics [8].

Subinhibitory antibiotics, such as azithromycin, cefamandole, ciprofloxacin, clarithromycin, erythromycin, linezolid, ofloxacin, rifampicin, sparfloxacin, tetracycline, and vancomycin, have been known to stimulate biofilm formation [9]. Studies have found that PIA genes were upregulated following treatment with sublethal tetracycline, erythromycin and quinupristin-dalfopristin [9]; however, they were not correlated with increased biofilm production after sublethal erythromycin, azithromycin, and clarithromycin exposure [10]. In addition, sublethal nafcillin and linezolid reduced the expression of *Staphylococcus aureus* virulence factors such as alpha-toxin, autolysin, coagulase, Panton-Valentine leukocidin, haemolysin, staphylococcal enterotoxin, and toxic-shock syndrome toxin [11]. This activity can lead to bacterial infection complications when biofilms are exposed to sub-minimum inhibitory concentration (MIC) concentrations of antibiotics.

Tigecycline (TC) belongs to the glycylcycline class and is a 9-glycylamido derivative of minocycline [12]. It has a broad spectrum of antibacterial effect against both Gram-positive and -negative bacteria, including methicillin-resistant staphylococci and tetracycline-resistant Enterobacteriaceae [13]. Because TC binds to the 30S ribosomal subunit with higher affinity than does tetracycline, it blocks tRNA from being delivered to the ribosomal A site, thereby impeding translation elongation [14].

Global proteomic analysis and bioinformatics have been employed to identify and characterize hundreds to thousands of differentially expressed proteins (DEPs) [15]. The Q-Exactive HF-X Orbitrap mass spectrometer is equipped with a high-capacity transmission tube that makes extremely sophisticated sensitivities possible, as well as an ultra-high field Orbitrap analyzer that increases resolving power up to 240,000 and scan speed up to 40 Hz [16]. Because a more comprehensive proteomic analysis can provide more direct insight into molecular responses, we employed a label-free quantitative proteome analysis using the Q-Exactive HF-X Orbitrap mass spectrometer to determine cellular response to environmental stresses [15]. The effect of subinhibitory concentrations of TC on protein expression in *S. epidermidis* strain RP62A biofilms using a proteomic technique was investigated.

## 2. Materials and Methods

### 2.1. Bacterial Strain and Growth Conditions

A single colony of *S. epidermidis* strain RP62A was inoculated in 1 mL of tryptic soy broth (TSB) supplemented with 0.25% glucose (TSBg) and cultured in a shaking incubator (200 rpm) (New Brunswick Scientific Co., New Brunswick, NJ, USA) at 37 °C overnight. The culture was diluted 1:200 in fresh TSBg and grown for 4 h. Bacterial cells were harvested by centrifugation at 20,817× *g* for 5 min at 4 °C and washed three times with phosphate-buffered saline (PBS). The bacterial suspension was adjusted to an optical density at 600 nm of 0.1 and grown with TSBg with 1/8 (0.031 µg/mL, T1), 1/4 (0.063 µg/mL, T2), and 1/2 (0.125 µg/mL, T3) MIC of TC (MilliporeSigma, St. Louis, MO, USA) or without TC in six-well plates (Corning Inc., Corning, NY, USA) with shaking at 100 rpm for 24 h at 37 °C. Next, planktonic cells were washed with PBS three times and biofilms were scraped using cell scrapers (Corning Inc.) and transferred to a microcentrifuge tube. The cells were centrifuged, washed with PBS, and stored at −80 °C before protein extraction.

### 2.2. Determination of Minimum Inhibitory Concentration (MIC) and Viable Bacterial Cell Numbers

To measure the MIC of TC, the antibiotic was serially diluted in Mueller-Hinton broth (MHB, Thermo Fisher Scientific, Waltham, MA, USA) in 96-well plates (Corning Inc.). *S. epidermidis* strain RP62A culture was adjusted to 0.5 McFarland standards and added to each well of the plate containing TC. The plate was placed in a Synergy 2 Multi-Mode Microplate Reader (BIOTEK Instruments, Winooski, VT, USA) at 37 °C and shaken continuously. Bacterial growth was monitored by reading the absorbance at 600 nm every 30 min for 24 h. The lowest concentration of antibiotic with no visible cell growth was defined as the MIC. All experiments were performed in quadruplicate. To measure the viable bacterial cell numbers as colony forming unit (CFU)/mL, biofilms were removed with a sterile cotton swab after washing biofilms with PBS, and suspended in 1 mL of PBS. They were vortexed for 30 s and sonicated at 40 kHz for 3 min. Ten-fold serial dilutions were performed to enumerate the viable cells on tryptic soy agar (TSA) plates in triplicate.

### 2.3. Confocal Laser Scanning Microscopy (CLSM)

Biofilm cells containing TSBg (control) or subinhibitory concentrations of TC were grown in a tissue culture treated 24 well plate (ibidi, Gräfelfing, Germany) overnight at 37 °C. They were washed with PBS three times and stained for 30 min in the dark at room temperature with the FilmTracer LIVE/DEAD Biofilm Viability Kit (Molecular Probes, Eugene, OR, USA) containing SYTO 9 and propidium iodide. After staining, the biofilms were gently rinsed with filter-sterilized water. Biofilm images were acquired with a Nikon Ti2 inverted microscope with an A1 HD25 confocal scan head, and images were processed using NIS Elements software (Nikon, Tokyo, Japan). Six locations within a well (one treatment group) were quasi-randomly selected at 20× magnification and then z-stack images (20 µm thickness, 0.5 µm slices) were collected using a Plan Flour 40× oil-corrected objective (NA = 1.3). The detection unit was fitted with GaAsP photomultiplier tubes for FITC and TRITC fluorescent signal acquisition. Parameters for the FITC laser were as follows: power = 5.0, gain = 40, and offset = 20. Parameters for the TRIC laser were as follows: power = 5.0, gain = 50, and offset = 20. Thresholding was used to calculate total areas for FITC and TRITC. The ratio of FITC (live cells) to TRITC (dead cells) was log-transformed and data were analyzed via one-way analysis of variance (ANOVA) with a Holm-Sidak correction for multiple comparison tests (α<0.05).

### 2.4. Field Emission Scanning Electron Microscopy (FESEM)

Biofilm cells were washed with PBS and dehydrated using 15%, 30%, 50%, 70%, 80%, 90%, 95%, and 100% ethanol. Next, samples were dried with hexamethyldisilazane (HMDS, MilliporeSigma) in 100% HMDS and a 1:2, 1:1, and 2:1 mixture of HMDS and ethanol. Finally, biofilms were sputter-coated with gold (Denton Vacuum, Moorestown, NJ, USA), and images were visualized using a Zeiss-Merlin FESEM (Carl Zeiss Microscopy, Thornwood, NY, USA).

### 2.5. Protein Extraction

Harvested biofilms were added to Lysing Matrix B tubes (MP Biomedicals, Santa Ana, CA, USA) containing 0.1 mm silica spheres. One hundred microliters of BugBuster Plus Lysonase kit (MilliporeSigma) was pipetted to the tube and the biofilms were disrupted by an FP120 reciprocator (MP Biomedicals) at speed 6 for 45 s. Disrupted cells were boiled and vortexed for 5 min and 1 min, respectively. The final protein extract was recovered by centrifuge at maximum speed at 4 °C.

### 2.6. Ultra-High Performance Liquid Chromatography-Tandem Mass Spectrometry (UHPLC-MS/MS)

Ultra-high performance liquid chromatography-tandem mass spectrometry (UHPLC-MS/MS) was conducted by Bioproximity, LLC (Manassas, VA, USA). The protein samples were suspended in 5% SDS, 50 mM Tris-HCl (pH 8.0), 5 mM Tris (2-carboxyethyl) phosphine (TCEP), and 20 mM 2-chloroacetamide. Protein digestion was achieved using the single-pot solid-phase-enhanced sample preparation (SP3) method [17]. Liquid chromatography (LC) was performed on an EASY-nLC 1200 (Thermo Fisher Scientific) connected to a Q-Exactive HF-X quadrupole-Orbitrap mass spectrometer (Thermo Fisher Scientific). The column was 25 cm × 50 μm I.D. and packed with 2 micron C18 media (Thermo Easy Spray PepMap, Thermo Fisher Scientific). The 20 min LC gradient ran from 10% B to 30% B over 16 min, to 45% B over 4 min, then to 80% B for the remaining 5 min. The mass spectrometer (MS) was set to acquire by data-dependent acquisition and tandem mass spectra from the top 12 ions in the full scan from *m/z* 350–1400. Normalized collision energy was set at 27, automatic gain control to 3e6, maximum fill MS to 45 ms, and maximum fill MS/MS to 22 ms. Data processing and library searching were performed as described by Burtnick et al. [18]. MS1-based label-free quantification was employed and peptide peak areas were calculated using OpenMS [19]. Proteins were required to have one or more unique peptides across the analyzed samples with E-value scores of 0.0001 or less. The cutoff between control and TC-treated groups was ≥2.0 (up) and ≤−2.0 (down). Functional annotation of the proteins was carried out by Cluster of Orthologous Groups (COG), and the Kyoto Encyclopedia of Genes and Genomes (KEGG) pathway analysis was employed for systematic analysis of gene function [20,21]. Protein interaction network analysis was performed using STRING database version 11.5 and Cytoscape version 3.9.1 [22,23]. Heatmap was generated using the R package ‘pheatmap’ (https://cran.r-project.org/web/packages/pheatmap/pheatmap.pdf) (accessed on 30 September 2022), and the protein expression pattern analysis was conducted in-house python script.

## 3. Results

### 3.1. MIC and Biofilm Morphology after TC Treatment

The MIC of TC in *S. epidermidis* strain RP62A was 0.25 µg/mL (Figure 1). Biofilms of strain RP62A treated with increasing concentrations of TC were imaged via CLSM (Figure 2). Visual inspection of z-stacks suggested that increasing cell death was associated with increasing TC concentration. Semi-quantitative image analysis of the ratio of live to dead bacteria via thresholding (NIS Elements) found a main effect between groups [F(3,23) = 14.831, *p* < 0.001]. However, multiple comparison tests found no difference among TC treatment groups (1/2 MIC vs. 1/4 MIC, *p* = 0.900; 1/2 vs. 1/8 MIC, *p* = 0.064; 1/4 vs. 1/8, *p* = 0.056; see Appendix A for details), but all TC treatment groups were different from control (all *p*’s < 0.01). This suggests that in the CLSM experiment, all concentrations of TC led to similar levels of bacterial death. An independent bacterial colony forming unit (CFU) experiment did find a relation between TC treatment and bacterial count (Figure 3).

FESEM with low magnification clearly showed that 1/8 and 1/4 MIC TC treatment promoted biofilm formation in strain RP62A (Figure 4A). In control biofilms, networks of filamentous structures surrounded the bacterial surface and entangled the bacteria (Figure 4B). Production of extracellular polymeric substances (EPS) was significantly less in the 1/2 MIC-treated biofilm than in the other biofilm samples. In both 1/8 and 1/4 MIC-treated biofilms, a massive, tangled, thread-like matrix covered the surface of the bacteria, making the cells hardly visible.

Some bacteria have been known to secrete membrane vesicles (MVs) that have round, bilayered, membranous structures from 25 to 250 nm [24]. MVs are composed of proteins, polysaccharides, nucleic acids, and lipids and carry several virulence factors and extracellular enzymes [25]. Particularly, they have been known to be an essential part of EPS and to help build biofilms. We found that *S. epidermidis* strain RP62A in each sample released MVs, indicated by red arrows in Figure 4B.

### 3.2. Global Overview of the Proteome Data

Distinct proteomic data sets were induced when strain RP62A biofilms were exposed to increasing levels of TC. From the proteomic data analysis, 484, 480, 530, and 629 proteins were identified from control, T1, T2, and T3 biofilms, respectively (Appendix A, Figure 5). A Venn diagram indicated that 303 proteins were commonly observed in all comparison groups, whereas 52, 8, 16, and 107 proteins were uniquely detected in control, T1, T2, and T3 biofilms, respectively (Figure 5). As showed in the heatmap analysis (Figure 6), the proteomic data set revealed an apparent correlation between the profiles of protein expression with respect to TC concentrations. In T1, T2, and T3 biofilms, 413, 429, and 518 proteins were differentially expressed, respectively, containing 165, 243, and 383 upregulated and 248, 186, and 135 downregulated (Figure 7).

According to COG functional category analysis [26], there are four main categories, including: (i) cellular processes and signaling, (ii) information storage and processing, (iii) metabolism, and (iv) poorly characterized. As the TC concentration increased, the number of upregulated proteins in each category increased, but the number of downregulated proteins decreased (Figure 8). This result indicates that strain RP62A biofilm cells sense and respond to TC in a different way, depending on the antibiotic concentration. Additionally, “information storage and processing” and “metabolism” were the most affected COG categories in upregulated proteins (Figure 8A).

When compared with the number of upregulated proteins, the distribution of the number of downregulated proteins was significantly different. The number (18 in T3 biofilm) of downregulated proteins in “information storage and processing” was significantly lower compared to the number (106 in T3 biofilm) of upregulated proteins (Figure 8B). The number (116 in T1 biofilm) of downregulated proteins was highest in “metabolism.” Apart from the “function unknown (S)” category of proteins, “translation, ribosomal structure and biogenesis (J)” class of proteins was identified as the most prevalent COG across upregulated proteins (Figure 9A). Again, apart from the “function unknown (S),” the main COGs categories of suppressed proteins functionally enriched were in “translation, ribosomal structure and biogenesis (J)”, “energy production and conversion (C), and “nucleotide transport and metabolism (F)”, as well as in “amino acid transport and metabolism (E)” (Figure 9B). Various functional protein groups in strain RP62A biofilms were differentially activated according to TC concentrations, a finding which could help predict differences in cellular phenotypes. A lot of upregulated and downregulated proteins belonged to “function unknown (S),” indicating there were difficulties in interpreting the proteome data.

As the TC concentration increased, the number of induced proteins in each KEGG category increased, but the number of repressed proteins decreased, a similar trend to that observed with COG data, indicating elevated dynamic cellular responses of biofilm cells to sub-MIC TC concentrations (Figure 10, Appendix A). Among five main KEGG categories including “metabolism,” “cellular processes,” “environmental information processing,” “genetic information processing,” and “human diseases,” the “metabolism” pathway accounted for most DEPs in both upregulated and downregulated proteins. Within the metabolism group, in addition to the “global and overview map” category, the “carbohydrate metabolism” and “amino acid metabolism” categories were the most populated in both upregulated and downregulated proteins (Figure 11, Appendix A). Within the two categories, a significant numbers of the differentially expressed proteins associated with glycolysis (39 proteins), the tricarboxylic acid (TCA) and pentose phosphate pathways (51 proteins), pyruvate oxidation (51 proteins), purine and pyrimidine metabolism (56 proteins), and amino acid and cofactor biosynthesis (219 proteins) were identified after TC treatment, indicating that these protein groups may play key roles assisting strain RP62A in defending against antibiotic pressure.

Protein expression pattern analysis tool that is integrated with COG functional category analysis, was used to cluster proteins that show a similar temporal expression pattern. In total, 27 significant temporal protein expression profiles were clustered as shown in Figure 12. When strain RP62A biofilms were treated with TC, about 100 proteins involved in information storage and processing (INF; J, K, and L), cellular processes and signaling (CEL; D, M, U, and O), and metabolism (MET; C, G, E, and P) were significantly upregulated (EPN22). Notably, translation, ribosomal structure and biogenesis (J) accounted for the largest portion (30 proteins, 40.5%) of the upregulated proteins. About 48 proteins included in EPN1 were consistently identified in all TC-treated biofilm samples including control and the majority of them consisted of translation, ribosomal structure and biogenesis (J), posttranslational modification, protein turnover, chaperones (O), and carbohydrate transport and metabolism (G). As revealed in EPN2, proteins associated with INF (J and L) and MET (C, G, E, F, H, I, P, and Q) were substantially downregulated in all TC-treated biofilms (T1, T2, and T3). Protein expression patterns that were upregulated (EPN19) or downregulated (EPN9) only at the highest TC concentration were confirmed.

The “ribosome” (upregulated and downregulated protein numbers, 37 and 0) and “ATP-binding cassette (ABC) transporter” (upregulated and downregulated protein numbers, 10 and 4) pathways showed the largest differences between up- and downregulated proteins in 1/2 MIC-treated biofilms. Repressed “Glycolysis/gluconeogenesis” (upregulated and downregulated protein numbers, 0 and 15) and “butanoate metabolism” (upregulated and downregulated protein numbers, 0 and 11) pathways in 1/8 MIC-treated biofilm samples were the most notable pathways compared to those of overexpressed proteins.

### 3.3. Impact of TC on Biofilm-Associated Proteins

Sub-MIC TC concentrations changed expression of 12 proteins involved in biofilm formation in *S. epidermidis* strain RP62A (Table 1). Bhp, Ebh, IcaB, SarA, SrrA, SERP0719, SERP1316, and SERP1483 were substantially overexpressed in 0.125 µg/mL TC-treated biofilms. Interestingly, three cell wall surface anchor family proteins showed the greatest induction in T2 and T3 biofilm samples. At the highest concentration of TC, the protein expression levels of biofilm-associated proteins were not significantly different from that of the biofilms treated at the low concentration.

### 3.4. Impact of TC on Antibiotic Resistance-Associated Proteins

Of the 70 antibiotic resistance-associated proteins obtained from the Pathosystems Resource Integration Center (PATRIC) database, in *S. epidermidis* strain RP62A, 27 were differentially regulated [27] (Table 2). The highest number of antibiotic resistance proteins were altered to a large extent in biofilms grown under the conditions with the highest TC concentrations. Proteins involved in aminoglycosides (AacA-AphD, AphA, RpsL), antimicrobial peptides (DltA, DltC, LytH, SepA), β-lactams (Pbp, MecA), erythromycin (ErmA-1), fluoroquinolones (NorR), fusidic acid (FusA), glycopeptides (Ddl, TcaA), lipopeptides (RpoC, MprF), mupirocin (IleS), rifampicin (RpoB), and trimethoprim (FolA-2) resistance, as well as tetracycline (RpsJ) resistance, were differentially expressed. Several antibiotic resistant genes, including aminoglycosides, streptomycin, ribostamycin, and paromomycin, were overexpressed in *A. baumannii* grown in the presence of TC [28]. RpoB, RplF, RpsJ, and FusA proteins were increased in biofilms treated with all three concentrations of TC, and their fold-ratios increased as the antibiotic concentration increased. RpoC, TcaA, and NorA were not differentially expressed in T1 biofilms, but were differentially expressed in both T1 and T2 biofilms.

### 3.5. Impact of TC on Virulence- and Quorum Sensing-Associated Proteins

Expression levels of 18 virulence proteins were considerably changed during growth of strain RP62A biofilms with sublethal TC (Table 3). CcpA and NorA were not differentially expressed in T1 biofilms, but their expression was altered in both T1 and T2 biofilms. ClpB, ClpX, NagD, PurL, RecA, and Asd were overexpressed only in 1/2 MIC-treated biofilms; among them, PurL, RecA, and Asd exhibited the highest upregulation (fold ratio 1000). It has been previously reported that antibiotics can affect the expression of toxins (*tst, pvl*), enzymes (*capB, capC, capD, capF, capG*, and *cap8H*), regulatory proteins (*agr, sarA, saeP* and *rot*), and other virulence determinants in staphylococci [11,29,30]. We found a single toxin (SERP0738) and eight exoenzymes (ClpB, ClpC, ClpP, ClpX, FumC, Geh-2, Lip, and SepA) were significantly altered after TC exposure.

Quorum sensing systems in bacteria control a variety of cellular processes, which include the regulation of biofilm formation, antimicrobial resistance, and virulence [31]. TC treatment of strain RP62A biofilms resulted in significant changes in 11 proteins related to quorum sensing (Table 4). SERP0738 and GseA were overexpressed only in T1 and T2, while SecD and YidC2 were only overexpressed in T2 and T3. YajC and SERP1994 showed the greatest induction by 1/2 MIC TC. Regardless of TC concentration, levels of FFh and Hfq were greatly induced, while SecA1 production was severely repressed.

### 3.6. Impact of TC on ABC Transporter- and Protein Export-Associated Proteins

ABC transporters have been closely related to multidrug resistance in both Gram-positive and -negative bacteria [32]. Addition of TC to RP62A biofilms resulted in 13 differentially expressed proteins linked to ABC transporters (Table 5). In TC-treated biofilm samples, 10 proteins related to ABC transporters were upregulated, but only two were downregulated. Highly abundant proteins in all TC-treated biofilms included substrate binding proteins (SERP0099, SERP0491, SERP1777, ModA, SERP2005, SERP2286, SERP2383), but permease proteins (SERP0386, ModB) were drastically downregulated in all TC-treated biofilms.

Subinhibitory TC treatment against strain RP62A led to altered expression of seven proteins linked to protein export (Table 6). While SpsB (signal peptidase IB) and Ffh (signal recognition particle protein) were overexpressed in all TC-treated biofilms, SecA1 (preprotein translocase) was downregulated in all TC-treated biofilms. In addition, expression levels of SERP0024 (signal peptidase I), SecD (protein-export membrane protein), and YidC2 (putative membrane protein) were predominantly increased only in biofilms challenged with 1/4 and 1/8 MIC TC.

### 3.7. Impact of TC on Purines and Pyrimidines-Associated Proteins

The number of overexpressed proteins related to both purine and pyrimidine metabolism in strain RP62A biofilms increased with TC concentration, showing the highest number of upregulated proteins at the highest TC concentration (Table 7 and Table 8). Among the most highly upregulated proteins (fold ratio 1000), significantly more were involved in pyrimidine metabolism (CarA, CarB, PyrR, Tdk, Udk) than were involved in purine metabolism (Gmk, PurL) in T3 biofilms.

### 3.8. Impact of TC on Ribosome-Associated Proteins

The majority of proteins associated with “ribosome” were upregulated in all sub-MIC treated biofilms, and RpsO, RplT, and RpsM showed the greatest upregulation (fold ratio 1000) in all T1, T2, and T3 (Table 9). Sixteen proteins belonging to the Rps family of 30S ribosomal proteins, 18 proteins belonging to the Rpl family of 50S ribosomal proteins, and three proteins belonging to the Rpm family of 50S ribosomal proteins were expressed at various levels. Translation initiation factors infA and infC, which are essential for instigating translation, were significantly overexpressed. The level of expression of elongation factors G (FusA) also was highly increased. Interestingly, RpsR, a ribosomal protein S18, was not detected in T1 and T2 biofilms but only upregulated at a fold ratio of 1000 in the highest MIC-treated biofilms. Four proteins, RplY, RpsD, RpsE, and RplP, were distinctly downregulated in the lowest MIC-treated biofilms.

### 3.9. Impact of TC on Essential Gene-Associated Proteins

Among 212 essential proteins of *S. epidermidis* strain RP62A, 94 (44.3%) proteins were differentially expressed following TC treatment of its biofilms (Table 10). Overexpressed proteins in T1, T2, and T3 biofilms were 31, 48, and 64, respectively, suggesting that the higher the strength of external stress, the more essential proteins are needed. Most of the differentially expressed essential proteins belong to “Information storage and processing (J)”, followed by “nucleotide transport and metabolism (F)”, “coenzyme transport and metabolism (H)”, and transcription (K)” (Table 10).

## 4. Discussion

In this study, we investigated the concentration-dependent proteome response of *S. epidermidis* strain RP62A biofilms to TC. Our results show that the number of differentially altered proteins increased with increasing TC concentration. However, Wu et al. examined the proteomic response of *Pseudomonas aeruginosa* planktonic cells to sublethal levels of tobramycin and reported that a higher number of proteins were significantly expressed at the lowest concentration of the antibiotic [33]. The discrepancy between the two studies suggests that planktonic and biofilm cells differ in their responses to antibiotic treatment. A significant number of proteins were induced at the lowest TC concentration (0.031 µg/mL), signifying that strain PR62A biofilms are able to respond to low concentrations of TC, which explains how TC at low concentrations may induce adaptive responses. Of the F proteins, 23 were identified with the highest level of expression (fold ratio 1000) in all T1, T2, and T3 biofilms. In contrast, 52 proteins were detected with the lowest fold ratios (fold ratio −1000) in all sublethal TC-treated biofilms, with proteins belonging to the metabolism pathway accounting for 90.8%. These results indicate that metabolism is critical for the resistance of strain RP62A biofilms to TC treatment.

Eleven proteins involved in the “TCA cycle” pathway were upregulated in 1/2 MIC-treated biofilms. This contradicts previous findings in *Acinetobacter baumannii* planktonic cells, which reported that the “TCA cycle” pathway was repressed under TC treatment [28]. Among the genetic information processing group, the “translation” category was the most predominant KEGG group in both upregulated and downregulated proteins (Appendix A). Several researchers reported that the expression of translation-related proteins was significantly altered under the pressure of antibiotics that inhibit protein synthesis, demonstrating that translational regulation is directly associated with response to antibiotic stress [34,35,36].

Previous work indicated TC exposure to preformed biofilms of *Staphylococcus aureus* increased the level of expression of a fibronectin-binding protein, a collagen adhesin, a clumping factor, and a transcriptional regulator [11]. However, our proteome analysis revealed that only the transcriptional regulator (SarA) was induced in 0.031 and 0.125 µg/mL TC-treated biofilms. IcaADBC proteins of the intercellular adhesin locus synthesize β-1-6-linked N-acetyl-glucosamine (PNAG) [37]. *icaB* encodes a deacetylase required for the surface maintenance of PNAG and solid biofilm production in *S. epidermidis* [38]. In all TC-treated biofilm samples, IcaB showed the highest upregulation, indicating that TC is directly associated with IcaB expression in strain RP62A biofilms. However, IcaA, IcaC and IcaD proteins constituting the Ica operon were not identified in any of TC-treated biofilm samples. Moreover, RsbU, which was proposed as a regulator of *icaADBC* transcription, was also not detected [39]. These results agree with Smith et al., who found that TC pressure in *S. aureus* caused biofilm-associated genes to express at various levels [11]. Other studies also supported the idea that treatment with antimicrobial agents, including colistin, induced differential expression in intercellular adhesin genes [40,41,42,43].

Histidine metabolism is known to be an essential pathway in biofilm production [44]. It has been reported that genes/proteins contributing to biofilm formation in *Staphylococcus* species, *Mycobacterium* species, and *A. baumannii* were related to histidine metabolism [45,46,47]. Our proteome data showed that levels of putative aldehyde dehydrogenase (AldA, SERP1729) and NADH:flavin oxidoreductase/fumarate reductase (SERP2381) expressed in strain RP62A biofilms were altered under sublethal TC pressure.

Interestingly, several antibiotic resistance proteins, ErmA-1, FolA-2, Aac-aphD, Rho, MecA, IleS, and MprF exhibited the highest upregulation only in T3 biofilm. Construction of a functional protein–protein network using Cytoscape network analysis in DEPs revealed a single distinct cluster (Figure 13). In addition, it showed that transcriptional and translational proteins closely linked to the center may play a role in the functioning of various antibiotic resistance proteins. We found that β-lactam resistance proteins, such as MecA, Pbp1, Pbp2, and Pbp3, were highly induced following TC treatment. This result is inconsistent with previous findings that *A. baumannii* grown with subinhibitory TC remarkably suppressed β-lactam resistance genes [48,49]. It has been reported that TC treatment of *A. baumannii* and *P. aeruginosa* induced efflux pumps (*macB, mexXY, emrAB*) and tetracycline resistance genes (*tetA(B), tetR*), but these genes were not identified in our proteome data [34,49,50].

Sheldon and Heinrichs reported that the expression of ABC transporter lipoprotein SitC plays a role in iron uptake and survival of staphylococci in vivo [51]. SitC was repressed at a fold ratio of −2.55 by 0.125 µg/mL TC treatment. In the present study, TC treatment of strain RP62A biofilms differentially modulated known virulence factors important for host interaction. This may complicate bacterial infections in cases of inadequate treatment, where bacterial biofilms may be exposed to subinhibitory concentrations of antimicrobial agents.

TC treatment of strain RP62A biofilms exhibited differential expression in proteins involved in quorum sensing. In *P. aeruginosa*, subinhibitory tobramycin exposure not only promoted biofilm formation, but also increased expression of quorum sensing genes [52]. In contrast, silver nanoparticles and antimicrobial peptides inhibited both biofilm production and the expression of proteins related to quorum sensing systems [53,54]. Moreover, Zhao et al. found that sublethal erythromycin reduced biofilm formation by *Streptococcus suis* but resulted in differential expression levels of quorum sensing proteins [55].

Contrary to our proteome data, Hua et al. reported that both substrate binding and permease proteins associated with the uptake of amino acids, alkane sulphonate, and sulphate were downregulated in the presence of TC [49]. Differentially expressed ABC transporters resulting from antibiotic treatment showed that these proteins may play an important role in extrinsic stress responses. Since the functions of most ABC transporters in *S. epidermidis* strain RP62A are still unknown, whether their involvement in the antibiotic resistance mechanism should be further verified.

Purines and pyrimidines are essential for the synthesis of RNA and DNA necessary for bacterial cell growth [56]. A close link between purine/pyrimidine metabolism and antimicrobial stress has been reported [57,58,59]. Genes/proteins involved in purine production in *S. aureus* and *Enterobacter cloacae* were highly upregulated under vancomycin, rifampicin, and ZnO nanoparticle stresses [59,60,61]. Ning et al. reported that phenyllactic acid and lactic acid treatment in *Bacillus cereus* affect DNA and RNA biosynthesis by altering the expression levels of proteins involved in purine and pyrimidine metabolism [62]. The dynamic response of purine/pyrimidine metabolism-related proteins to increasing TC concentrations suggests that these proteins may modulate strain RP62A’s defense against the antibiotic pressure.

Most ribosomal proteins were upregulated in strain RP62A biofilms treated with subinhibitory TCs. It was previously reported that ribosomal proteins were differentially expressed following TC treatment on *S. aureus* biofilms [11]. In many bacteria, such as *A. baumannii, E. coli, Haemophilus influenzae*, and *Streptococcus pneumoniae*, overexpression of ribosomal proteins after antibiotic stress, which inhibits protein synthesis, was also reported [35,49,63,64]. These results suggest RP62A biofilms may reduce the effect of TC by increasing the proteins involved in ribosomal biosynthesis.

The “Information storage and processing (J)” protein set in essential genes is comprised of 41 ribosomal proteins and 13 aminoacyl-tRNA biosynthesis proteins. Differential expression of aminoacyl-tRNA biosynthesis genes was also reported in *A. baumannii* and *S. pneumoniae* following treatment with chloramphenicol, tetracycline, or TC [35,48]. Notably, RpsO (30S ribosomal protein S15), RplT (50S ribosomal protein L20), and RpsM (30S ribosomal protein S13) were overexpressed at the highest level in all TC-treated biofilms, demonstrating that these proteins have vital functions in biofilm survival during TC treatment. Cytoscape network analysis visually demonstrated the intimate interaction of ribosomal proteins and aminoacyl-tRNA biosynthesis proteins (Figure 14). The observed changes in protein levels support that many of these essential genes are not only essential for the growth of RP62A biofilms, but also have integral functions for survival against TC stress.

## 5. Conclusions

This study demonstrated that as TC concentration increased, the number of living cells in *S. epidermidis* strain RP62A biofilms decreased, but the biofilm densities varied. In addition, we found the global proteome response in strain RP62A biofilms to be dependent on TC concentration, with differential expression of diverse functional groups of proteins—including biofilm formation, antimicrobial resistance, virulence, quorum sensing, ABC transporters, and protein export—identified during treatment with different sub-MIC levels of TC. Biofilm cells exposed to subinhibitory concentrations of TC also exhibited active upregulation of proteins involved in metabolic pathways. Further studies of the molecular mechanisms by which functional proteins operate in biofilm protection in response to sublethal antibiotic stress need to be conducted.

## Figures and Tables

**Figure 1 cells-11-03488-f001:**
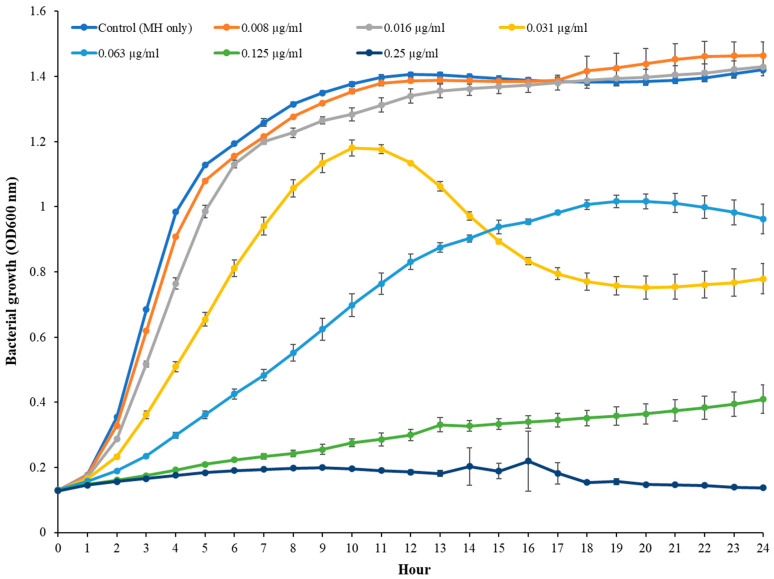
MIC measurement of TC in *S. epidermidis* RP62A.

**Figure 2 cells-11-03488-f002:**
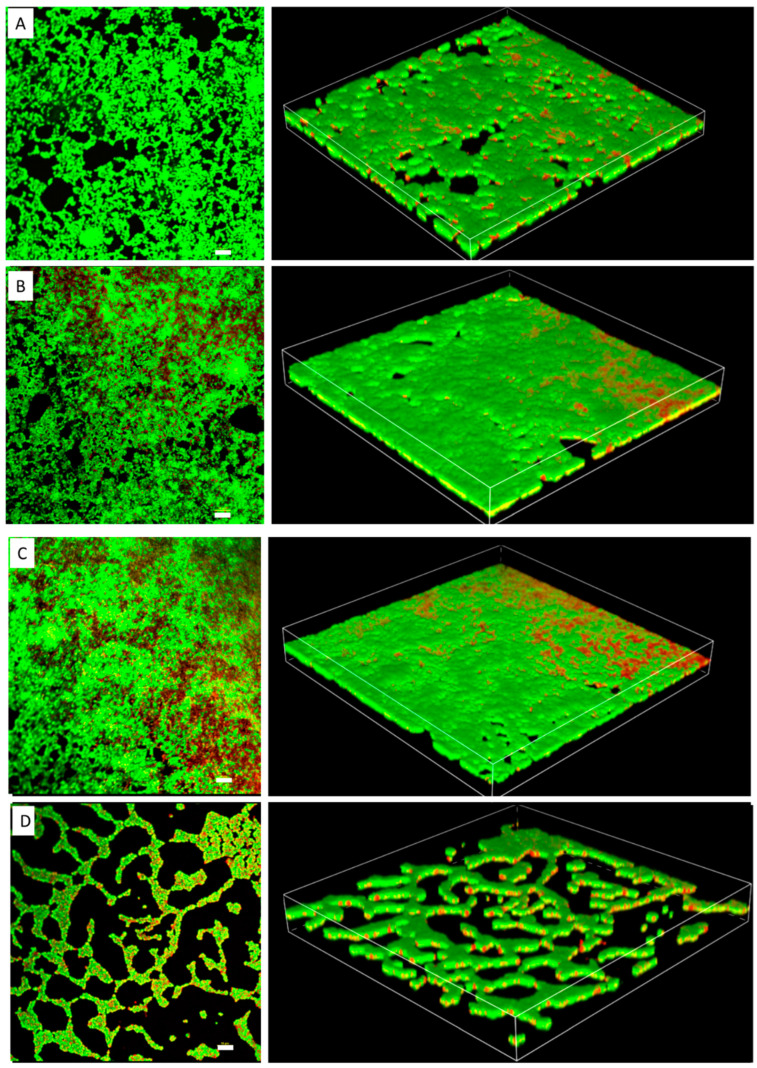
Confocal laser scanning microscopy images of *S. epidermidis* strain RP62A biofilms. The scale bar in all the images corresponds to 10 μm. Left panels show an orthogonal view of the top biofilm layer and right panels show 3D images. (**A**): Control, (**B**): 1/8 MIC, (**C**): 1/4 MIC, (**D**): 1/2 MIC.

**Figure 3 cells-11-03488-f003:**
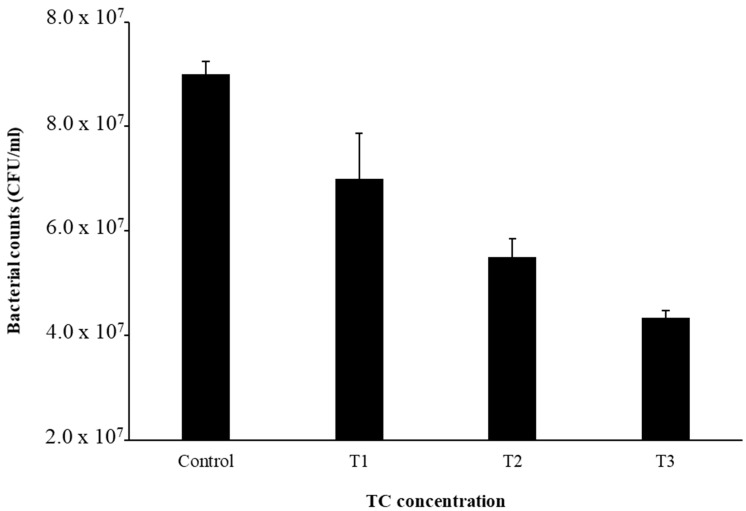
Bacterial cell counts after TC treatment. Experiments were performed in triplicate to calculate the mean and standard deviation.

**Figure 4 cells-11-03488-f004:**
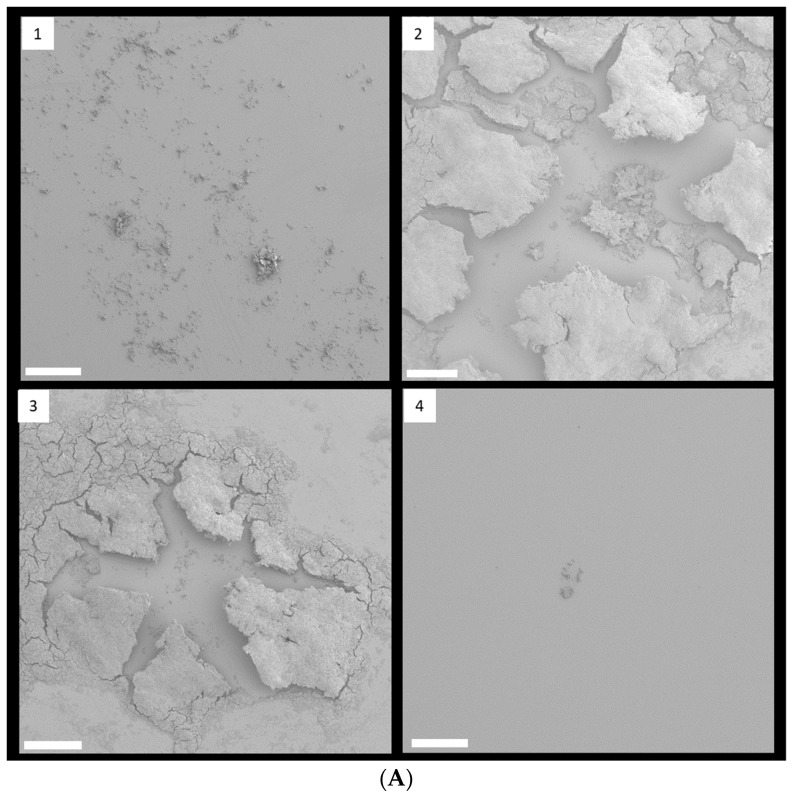
FESEM images of *S. epidermidis* strain RP62A biofilms. The scale bar in all the images corresponds to 200 µm (**A**) and 0.5 µm (**B**). Arrows indicate membrane vesicles. (**1**): Control, (**2**): 1/8 MIC, (**3**): 1/4 MIC, (**4**): 1/2 MIC.

**Figure 5 cells-11-03488-f005:**
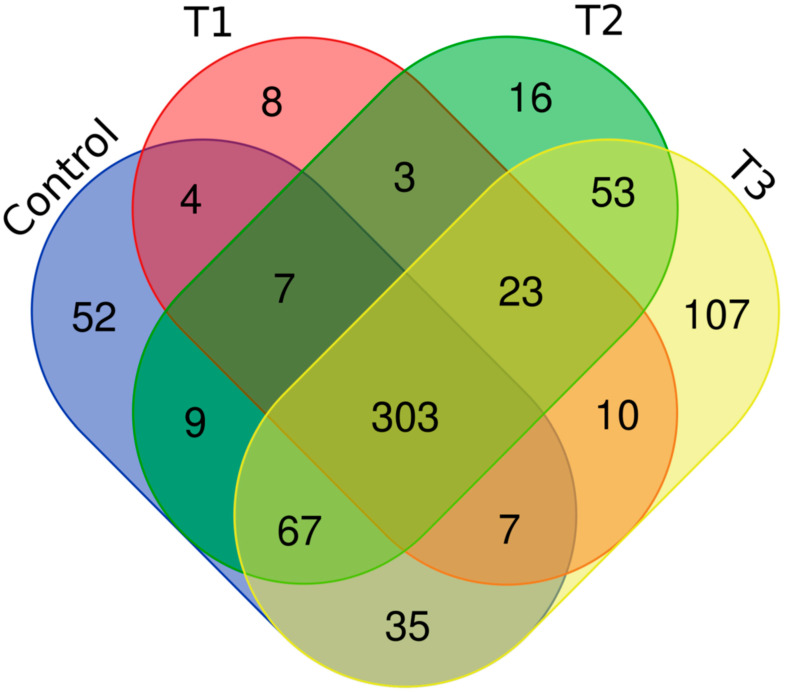
Venn diagram of the proteomic data in *S. epidermidis* strain RP62A biofilms. T1: 1/8 MIC, T2: 1/4 MIC, T3: 1/2 MIC.

**Figure 6 cells-11-03488-f006:**
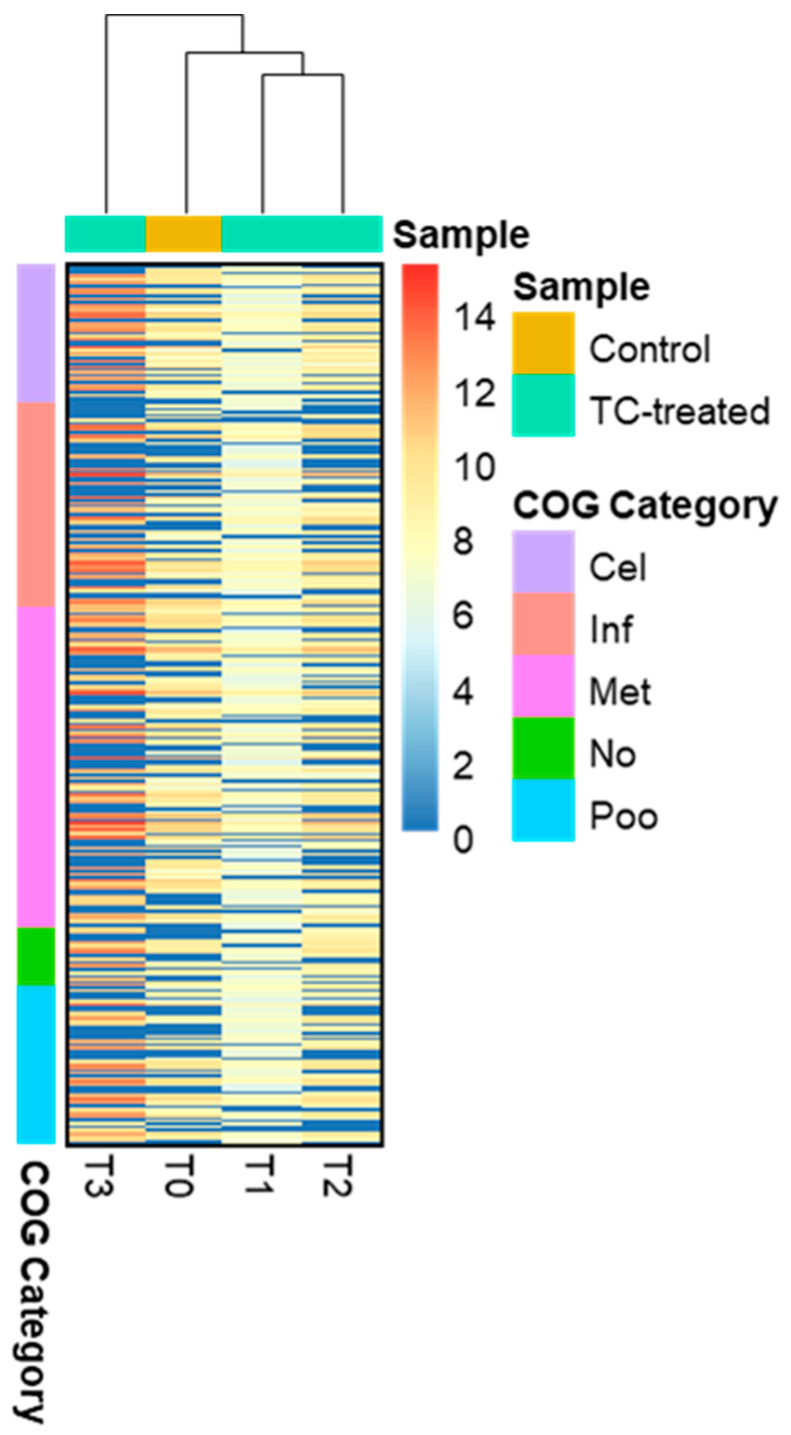
Heatmap presenting differentially expressed proteins. The proteins in the heatmap analysis were clustered according to five COG functional groups; Cel (Cellular process and signaling), Inf (Information storage and processing), Met (Metabolism), No (No COG annotation), and Poo (Poorly characterized).

**Figure 7 cells-11-03488-f007:**
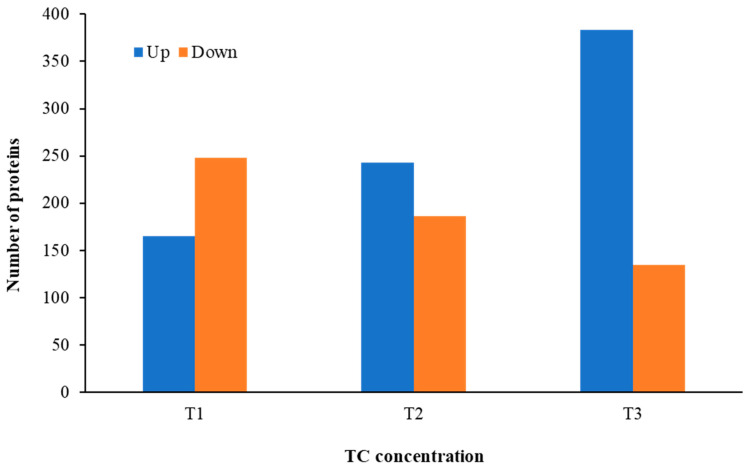
The number of differentially expressed proteins identified from *S. epidermidis* strain RP62A biofilms grown with sublethal MIC of TC. T1: 1/8 MIC, T2: 1/4 MIC, T3: 1/2 MIC.

**Figure 8 cells-11-03488-f008:**
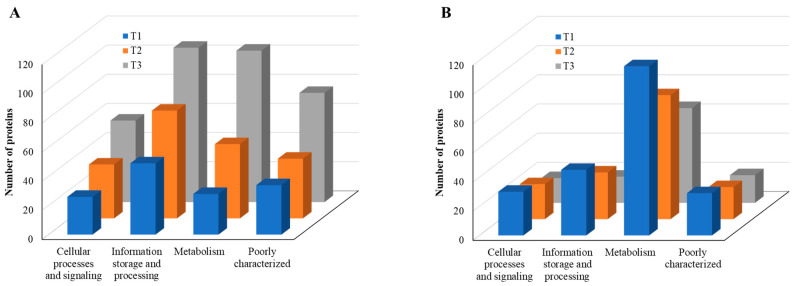
COG functional classification of four major categories in differentially expressed proteins identified from *S. epidermidis* strain RP62A biofilms grown with sublethal MIC of TC. (**A**): upregulated proteins, (**B**): downregulated proteins, T1: 1/8 MIC, T2: 1/4 MIC, T3: 1/2 MIC.

**Figure 9 cells-11-03488-f009:**
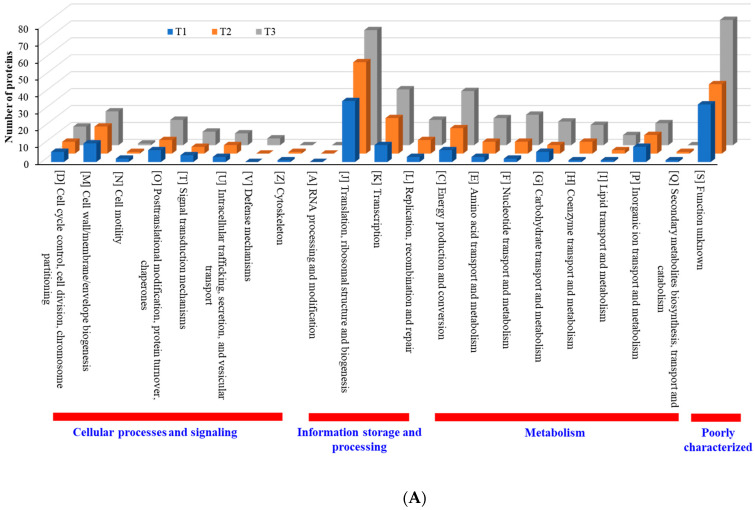
Detailed COG functional classification in differentially expressed proteins identified from *S. epidermidis* strain RP62A biofilms grown with sublethal MIC of TC. (**A**): upregulated proteins, (**B**): downregulated proteins, T1: 1/8 MIC, T2: 1/4 MIC, T3: 1/2 MIC.

**Figure 10 cells-11-03488-f010:**
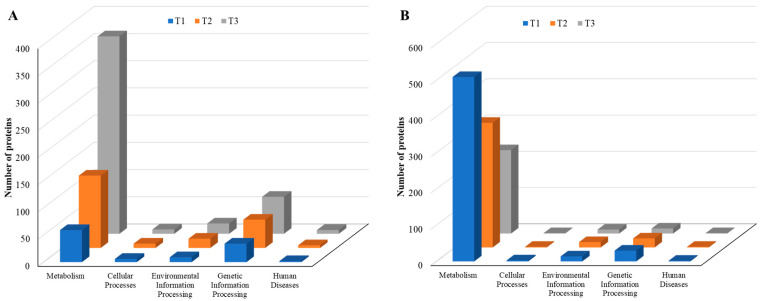
KEGG pathways of five major groups in differentially expressed proteins identified from *S. epidermidis* strain RP62A biofilms grown with sublethal MIC of TC. (**A**): upregulated proteins, (**B**): downregulated proteins, T1: 1/8 MIC, T2: 1/4 MIC, T3: 1/2 MIC.

**Figure 11 cells-11-03488-f011:**
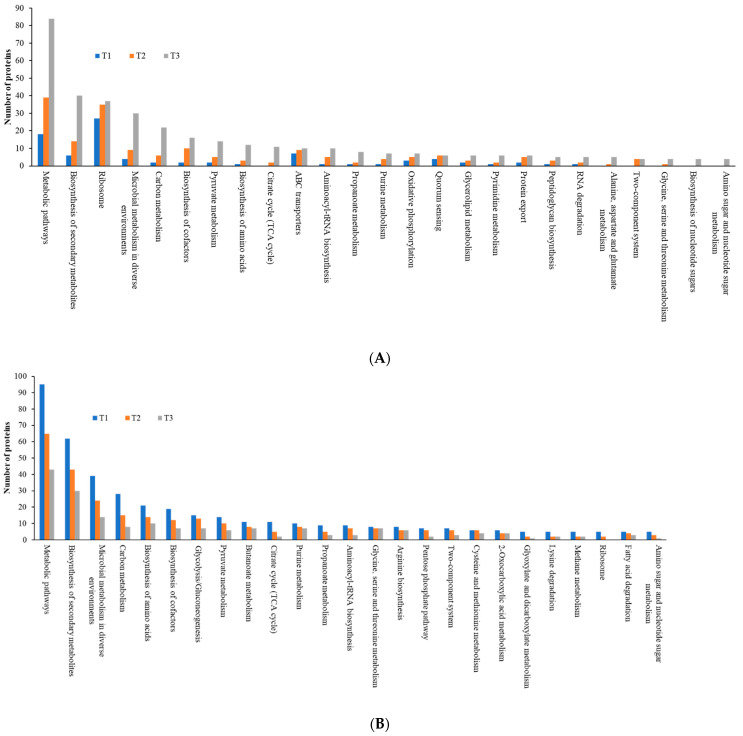
Top 25 KEGG pathways in differentially expressed proteins identified from *S. epidermidis* strain RP62A biofilms grown with sublethal MIC of TC. (**A**): upregulated proteins, (**B**): downregulated proteins, T1: 1/8 MIC, T2: 1/4 MIC, T3: 1/2 MIC.

**Figure 12 cells-11-03488-f012:**
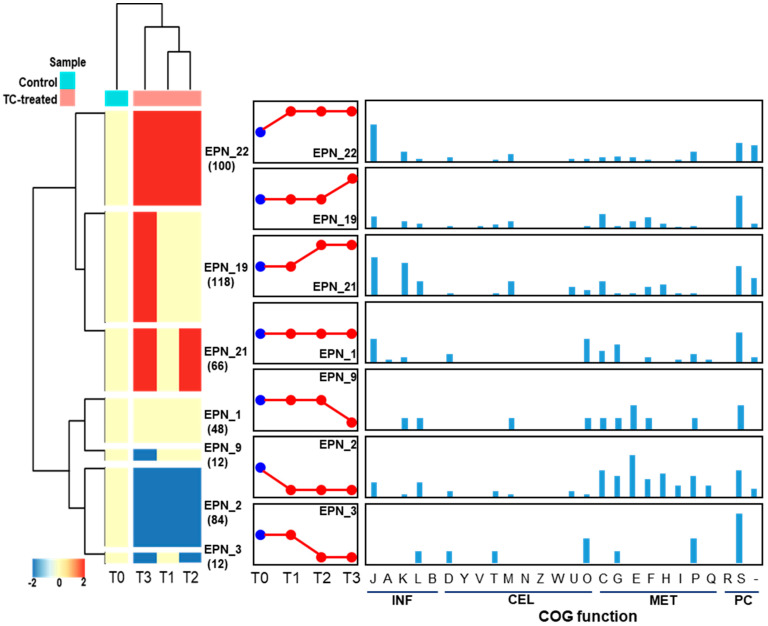
Protein expression patterns and functional distribution of proteins showing TC-dependent expression changes. INF, information storage and processing; CEL, cellular processes and signaling; MET, metabolism; PC, poorly characterized. COG functional categories: J, translation, ribosomal structure, and biogenesis; A, RNA processing and modification; K, transcription; L, replication, recombination, and repair; B, chromatin structure and dynamics; D, cell cycle control, cell division, chromosome partitioning; Y, nuclear structure; V, defense mechanisms; T, signal transduction mechanisms; M, cell wall/membrane/envelope biogenesis; N, cell motility; Z, cytoskeleton; W, extracellular structures; U, intracellular trafficking, secretion, and vesicular transport; O, posttranslational modification, protein turnover, chaperones; C, energy production and conversion; G, carbohydrate transport and metabolism; E, amino acid transport and metabolism; F, nucleotide transport and metabolism; H, coenzyme transport and metabolism; I, lipid transport and metabolism; P, inorganic ion transport and metabolism; Q, secondary metabolite biosynthesis, transport, and catabolism; R, general function prediction only; S, function unknown.

**Figure 13 cells-11-03488-f013:**
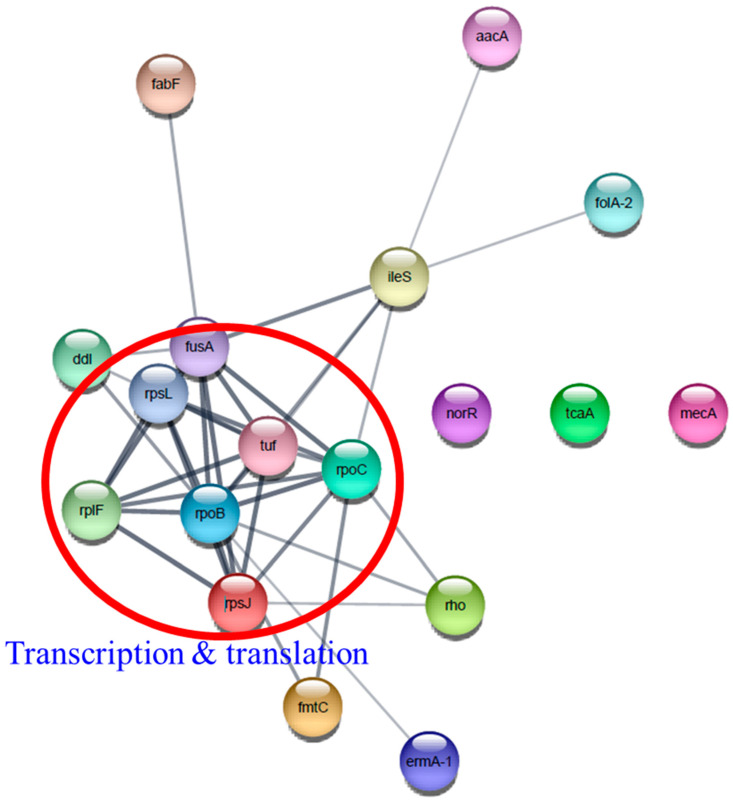
Protein–protein interaction network analysis of differentially expressed proteins associated with antibiotic resistance using STRING database and Cytoscape. Transcription and translation proteins were indicated by a red circle.

**Figure 14 cells-11-03488-f014:**
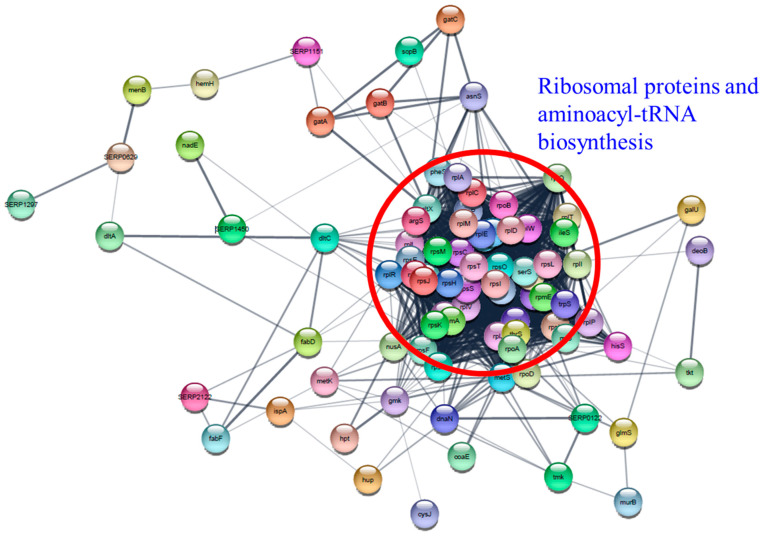
Protein–protein interaction network analysis of differentially expressed proteins associated with essential genes using STRING database and Cytoscape. Ribosomal proteins and aminoacyl-tRNA biosynthesis proteins were indicated by a red circle.

**Table 1 cells-11-03488-t001:** Differentially expressed proteins associated with biofilms.

					Fold Ratio		
Locus Tag	Gene	PATRIC ID	COG	Description	T1	T2	T3
SERP2398	aap	fig|176279.9.peg.2329	DM	Accumulation associated protein	5.97	1.82	−1.73
SERP0636	atl	fig|176279.9.peg.617	M	Peptidoglycan hydrolase	3.34	2.83	−1.40
SERP2392	bhp	fig|176279.9.peg.2323	O	Cell wall associated biofilm protein	−7.05	−52.76	3.08
SERP1011	ebh	fig|176279.9.peg.985	D	Extracellular matrix-binding protein	−1000.00	7.04	15.03
SERP2295	icaB	fig|176279.9.peg.2230	G	Poly-beta-1,6-N-acetyl-D-glucosamine N-deacetylase	1000.00	1000.00	1000.00
SERP1487	sdrH	fig|176279.9.peg.1456	DZ	Serine-aspartate repeat protein (surface protein)	1000.00	1000.00	0.00
SERP1316	SERP1316	fig|176279.9.peg.1283	D	Cell wall surface anchor family protein	0.00	1000.00	1000.00
SERP0719	SERP0719	fig|176279.9.peg.698	S	Cell wall surface anchor family protein	0.00	1000.00	1000.00
SERP1483	SERP1483	fig|176279.9.peg.1452	-	Cell wall surface anchor family protein	0.00	1000.00	1000.00
SERP2264	SERP2264	fig|176279.9.peg.2202	-	Cell wall surface anchor family protein	8.87	16.45	1.69
SERP0274	sarA	fig|176279.9.peg.261	K	Staphylococcal accessory regulator A	1000.00	0.00	1000.00
SERP1055	srrA	fig|176279.9.peg.1028	T	DNA-binding response regulator	−18.19	−2.68	3.32

**Table 2 cells-11-03488-t002:** Differentially expressed proteins associated with antibiotic resistance.

					Fold Ratio		
Locus Tag	Gene	PATRIC ID	COG	Description	T1	T2	T3
SERP1220	ermA-1	fig|176279.9.peg.1187	J	Erythromycin resistance protein	0.00	0.00	1000.00
SERP1581	folA-2	fig|176279.9.peg.1546	H	Dihydrofolate reductase	0.00	0.00	1000.00
SERP1585	aacA-aphD	fig|176279.9.peg.1550	F	Bifunctional AAC/APH	0.00	0.00	1000.00
SERP1690	ddl	fig|176279.9.peg.1650	F	D-alanine--D-alanine ligase	−1000.00	−1000.00	−1000.00
SERP1728	rho	fig|176279.9.peg.1687	K	Transcription termination factor	0.00	0.00	1000.00
SERP0183	rpoB	fig|176279.9.peg.174	K	DNA-directed RNA polymerase subunit beta	16.65	9.65	267.97
SERP0184	rpoC	fig|176279.9.peg.175	K	DNA-directed RNA polymerase subunit beta	0.00	1000.00	1000.00
SERP0186	rpsL	fig|176279.9.peg.177	J	30S ribosomal protein S12	1000.00	0.00	1000.00
SERP1816	rplF	fig|176279.9.peg.1773	J	50S ribosomal protein L6	26.76	36.00	49.69
SERP1832	rpsJ	fig|176279.9.peg.1789	J	30S ribosomal protein S10	2.53	4.45	15.72
SERP0188	fusA	fig|176279.9.peg.179	J	Elongation factor G	3.29	6.15	8.75
SERP0189	tuf	fig|176279.9.peg.180	J	Elongation factor Tu	−6.12	−2.74	−1.25
SERP1948	tcaA	fig|176279.9.peg.1897	S	Membrane-associated protein	0.00	1000.00	1000.00
SERP2521	mecA	fig|176279.9.peg.2441	M	Penicillin-binding protein 2	0.00	0.00	1000.00
SEA0010	aphA	fig|176279.9.peg.2480	J	Aminoglycoside 3′-phosphotransferase	−5.27	−1000.00	1.94
SERP0342	norR	fig|176279.9.peg.326	K	Transcriptional regulator, MarR family	1.06	3.78	5.59
SERP0568	fabF	fig|176279.9.peg.549	I	3-oxoacyl-[acyl-carrier-protein] synthase 2	−16.63	−1.86	1.00
SERP0746	pbp1	fig|176279.9.peg.725	M	Penicillin-binding protein 1	1000.00	1000.00	1000.00
SERP1020	pbp2	fig|176279.9.peg.994	M	Penicillin-binding protein 2	1.45	8.42	16.25
SERP1117	pbp3	fig|176279.9.peg.1088	M	Penicillin-binding protein 3	0.00	1000.00	1000.00
SERP0311	SERP0311	fig|176279.9.peg.295	GM	Antimicrobial peptide sensor	−1000.00	−1000.00	1.15
SERP0758	ileS	fig|176279.9.peg.737	J	Isoleucine--tRNA ligase	0.00	0.00	1000.00
SERP0518	dltA	fig|176279.9.peg.496	Q	D-alanine--D-alanyl carrier protein ligase	0.00	1000.00	0.00
SERP0520	dltC	fig|176279.9.peg.498	IQ	D-alanyl carrier protein	−1000.00	−106.80	−1.94
SERP1194	lytH	fig|176279.9.peg.1163	M	Probable cell wall amidase	0.00	0.00	1000.00
SERP2252	sepA	fig|176279.9.peg.2190	E	Neutral metalloproteinase	1000.00	1000.00	1000.00
SERP0930	mprF	fig|176279.9.peg.907	S	Phosphatidylglycerol lysyltransferase	0.00	0.00	1000.00

**Table 3 cells-11-03488-t003:** Differentially expressed proteins associated with virulence.

					Fold Ratio		
Locus Tag	Gene	PATRIC ID	COG	Description	T1	T2	T3
SERP0564	clpB	fig|176279.9.peg.545	O	Chaperone protein	−1000.00	−1.46	3.85
SERP0165	clpC	fig|176279.9.peg.156	O	ATP-dependent Clp protease ATP-binding subunit	6.85	4.58	5.69
SERP0436	clpP	fig|176279.9.peg.415	OU	ATP-dependent Clp protease proteolytic subunit	−3.67	−1.81	−2.11
SERP1238	clpX	fig|176279.9.peg.1205	O	ATP-dependent Clp protease ATP-binding subunit	−2.08	1.22	2.98
SERP1296	ccpA	fig|176279.9.peg.1264	K	Catabolite control protein A	−1.02	5.43	40.64
SERP0290	sitC	fig|176279.9.peg.275	P	ABC transporter, substrate-binding protein	1.25	−1.28	−2.55
SERP0342	norR	fig|176279.9.peg.326	K	Transcriptional regulator, MarR family	1.06	3.78	5.59
SERP1387	fumC	fig|176279.9.peg.1355	C	Fumarate hydratase class II	−1000.00	19.99	5.65
SERP2297	lip	fig|176279.9.peg.2232	D	Lipase	−1000.00	−1000.00	−1000.00
SERP2388	geh-2	fig|176279.9.peg.2320	D	Triacylglycerol lipase	15.08	108.01	7.98
SERP2252	sepA	fig|176279.9.peg.2190	E	Neutral metalloproteinase	1000.00	1000.00	1000.00
SERP0515	nagD	fig|176279.9.peg.493	G	Acid sugar phosphatase	−1000.00	−1.04	5.28
SERP0654	purL	fig|176279.9.peg.634	F	Phosphoribosylformylglycinamidine synthase subunit	0.00	0.00	1000.00
SERP0768	carA	fig|176279.9.peg.746	F	Carbamoyl-phosphate synthase small chain	0.00	1000.00	1000.00
SERP0738	SERP0738	fig|176279.9.peg.717	S	Phenol soluble modulin β1	1000.00	1000.00	0.00
SERP0852	recA	fig|176279.9.peg.831	L	Recombinase A	0.00	0.00	1000.00
SERP0921	acnA	fig|176279.9.peg.898	C	Aconitate hydratase A	−1000.00	−1.25	1.16
SERP0964	asd	fig|176279.9.peg.942	E	Aspartate-semialdehyde dehydrogenase	0.00	0.00	1000.00

**Table 4 cells-11-03488-t004:** Differentially expressed proteins associated with quorum sensing.

					Fold Ratio		
Locus Tag	Gene	PATRIC ID	COG	Description	T1	T2	T3
SERP0420	secA1	fig|176279.9.peg.400	U	Protein translocase subunit	−1000.00	−2.93	−1000.00
SERP0738	SERP0738	fig|176279.9.peg.717	S	Phenol soluble modulin beta 1	1000.00	1000.00	0.00
SERP0803	ffh	fig|176279.9.peg.781	U	Signal recognition particle protein	1000.00	1000.00	1000.00
SERP0871	hfq	fig|176279.9.peg.850	J	RNA-binding protein	1000.00	1000.00	1000.00
SERP1201	secD	fig|176279.9.peg.1170	U	Multifunctional fusion protein	0.00	1000.00	1000.00
SERP1202	yajC	fig|176279.9.peg.1171	U	Preprotein translocase	−1000.00	1.50	2.70
SERP1322	rot	fig|176279.9.peg.1289	K	HTH-type transcriptional regulator	−1000.00	−1.66	−1.43
SERP1397	gseA	fig|176279.9.peg.1366	M	Glutamyl endopeptidase	6.76	6.35	−1.20
SERP1697	yidC2	fig|176279.9.peg.1657	U	Membrane protein insertase	0.00	1000.00	1000.00
SERP1741	luxS	fig|176279.9.peg.1700	H	S-ribosylhomocysteine lyase	1.29	−2.85	−1.60
SERP1994	SERP1994	fig|176279.9.peg.1940	E	ABC transporter, substrate-binding protein	0.00	0.00	1000.00

**Table 5 cells-11-03488-t005:** Differentially expressed proteins associated with ABC transporter pathway.

					Fold Ratio		
Locus Tag	Gene	PATRIC ID	COG	Description	T1	T2	T3
SERP0099	SERP0099	fig|176279.9.peg.92	P	Lipoprotein	2.10	3.66	6.04
SERP0386	SERP0386	fig|176279.9.peg.368	P	ABC transporter, permease protein	−1000.00	−1000.00	−1000.00
SERP0491	SERP0491	fig|176279.9.peg.470	M	Lipoprotein	2.56	2.87	2.90
SERP1777	SERP1777	fig|176279.9.peg.1733	P	Iron compound ABC transporter	22.55	19.37	10.98
SERP1860	modB	fig|176279.9.peg.1817	P	Molybdenum transport system permease	−1000.00	−1000.00	−1000.00
SERP1861	modA	fig|176279.9.peg.1818	P	Molybdenum ABC transporter	2.46	3.82	3.63
SERP2005	SERP2005	fig|176279.9.peg.1953	ET	Amino acid ABC transporter	3.74	2.84	3.58
SERP2029	SERP2029	fig|176279.9.peg.1976	M	Amino acid ABC transporter	0.00	1000.00	1000.00
SERP2031	SERP2031	fig|176279.9.peg.1978	P	Amino acid ABC transporter	0.00	1000.00	1000.00
SERP2101	rbsD	fig|176279.9.peg.2044	G	D-ribose pyranase	0.00	0.00	1000.00
SERP2286	SERP2286	fig|176279.9.peg.2223	P	Phosphonate ABC transporter	6.26	10.52	7.04
SERP0290	SitC	fig|176279.9.peg.275	P	ABC transporter, substrate-binding protein	1.25	−1.28	−2.55
SERP2383	SERP2383	fig|176279.9.peg.2314	P	ABC transporter, substrate-binding protein	5.15	107.65	83.86

**Table 6 cells-11-03488-t006:** Differentially expressed proteins associated with protein export pathway.

					Fold Ratio		
Locus Tag	Gene	PATRIC ID	COG	Description	T1	T2	T3
SERP0024	SERP0024	fig|176279.9.peg.23	U	Signal peptidase I	0.00	1000.00	1000.00
SERP0420	secA1	fig|176279.9.peg.400	U	Protein translocase subunit	−1000.00	−2.93	−1000.00
SERP0553	spsB	fig|176279.9.peg.533	U	Signal peptidase I	3.77	12.13	14.23
SERP0803	ffh	fig|176279.9.peg.781	U	Signal recognition particle protein	1000.00	1000.00	1000.00
SERP1201	secD	fig|176279.9.peg.1170	U	Multifunctional fusion protein	0.00	1000.00	1000.00
SERP1202	yajC	fig|176279.9.peg.1171	U	Preprotein translocase	−1000.00	1.50	2.70
SERP1697	yidC2	fig|176279.9.peg.1657	U	Membrane protein insertase	0.00	1000.00	1000.00

**Table 7 cells-11-03488-t007:** Differentially expressed proteins associated with purine metabolism.

					Fold Ratio		
Locus Tag	Gene	PATRIC_ID	COG	Description	T1	T2	T3
SERP2186	sat	fig|176279.9.peg.2128	H	Sulfate adenylyltransferase	−1000.00	−1000.00	−1000.00
SERP2352	arcC	fig|176279.9.peg.2284	E	Carbamate kinase	−4.17	−13.57	−11.81
SERP0652	purS	fig|176279.9.peg.632	F	Phosphoribosylformylglycinamidine synthase	−1.29	−1.50	−10.02
SERP0657	purN	fig|176279.9.peg.637	F	Phosphoribosylglycinamide formyltransferase	−1000.00	−8.98	−5.48
SERP1870	ureB	fig|176279.9.peg.1827	E	Urease, β subunit	−1.53	−1.96	−4.50
SERP1743	deoB	fig|176279.9.peg.1702	G	Phosphopentomutase	−19.11	−4.55	−2.66
SERP0653	purQ	fig|176279.9.peg.633	F	Phosphoribosylformylglycinamidine synthase I	−1000.00	−1000.00	−2.00
SERP0656	purM	fig|176279.9.peg.636	F	Phosphoribosylformylglycinamidine cyclo-ligase	−1000.00	−1000.00	−1.81
SERP0651	purC	fig|176279.9.peg.631	F	Phosphoribosylaminoimidazole-succinocarboxamide synthase	1.76	4.61	−1.75
SERP0069	guaB	fig|176279.9.peg.63	F	Inosine-5′-monophosphate dehydrogenase	−8.04	−2.66	1.17
SERP0149	hpt	fig|176279.9.peg.140	F	Hypoxanthine phosphoribosyltransferase	−4.04	1.27	1.97
SERP1810	adk	fig|176279.9.peg.1767	F	Adenylate kinase	2.04	2.50	2.89
SERP0734	SERP0734	fig|176279.9.peg.713	F	Ham1 family protein	−1.79	1.61	3.03
SERP1869	ureA	fig|176279.9.peg.1826	E	Urease, γ subunit	−1000.00	3.44	3.03
SERP0067	xpt	fig|176279.9.peg.61	F	Xanthine phosphoribosyltransferase	−1.42	2.63	3.13
SERP0070	guaA	fig|176279.9.peg.64	F	GMP synthase	−1000.00	−1000.00	9.91
SERP0654	purL	fig|176279.9.peg.634	F	Phosphoribosylformylglycinamidine synthase II	0.00	0.00	1000.00
SERP0776	gmk	fig|176279.9.peg.754	F	Guanylate kinase	0.00	0.00	1000.00

**Table 8 cells-11-03488-t008:** Differentially expressed proteins associated with pyrimidine metabolism.

					Fold Ratio		
Locus Tag	Gene	PATRIC ID	COG	Description	T1	T2	T3
SERP0120	tmk	fig|176279.9.peg.112	F	Thymidylate kinase	−1000.00	−1000.00	−1000.00
SERP0764	pyrR	fig|176279.9.peg.742	F	Bifunctional protein	0.00	0.00	1000.00
SERP0767	SERP0767	fig|176279.9.peg.745	F	Dihydroorotase	0.00	1000.00	0.00
SERP0768	carA	fig|176279.9.peg.746	F	Carbamoyl-phosphate synthase small chain	0.00	1000.00	1000.00
SERP0769	carB	fig|176279.9.peg.747	F	Carbamoyl-phosphate synthase large chain	0.00	0.00	1000.00
SERP0771	pyrE	fig|176279.9.peg.749	F	Orotate phosphoribosyltransferase	−1000.00	−1.21	1.15
SERP0825	pyrH	fig|176279.9.peg.804	F	Uridylate kinase	−1000.00	−1000.00	−1000.00
SERP1045	cmk	fig|176279.9.peg.1019	F	Cytidylate kinase	−1000.00	−9.51	1.58
SERP1175	udk	fig|176279.9.peg.1144	F	Uridine kinase	0.00	0.00	1000.00
SERP1718	upp	fig|176279.9.peg.1678	F	Uracil phosphoribosyltransferase	4.11	−1.17	13.59
SERP1726	tdk	fig|176279.9.peg.1685	F	Thymidine kinase	0.00	0.00	1000.00
SERP1744	pdp	fig|176279.9.peg.1703	F	Pyrimidine-nucleoside phosphorylase	−6.23	−1.87	−1.33

**Table 9 cells-11-03488-t009:** Differentially expressed proteins associated with ribosome pathway.

					Fold Ratio		
Locus Tag	Gene	PATRIC ID	COG	Description	T1	T2	T3
SERP0046	rpsR	fig|176279.9.peg.45	J	30S ribosomal protein S18	0.00	0.00	1000.00
SERP0139	rplY	fig|176279.9.peg.131	J	50S ribosomal protein L25	−2.10	2.98	8.57
SERP0179	rplA	fig|176279.9.peg.170	J	50S ribosomal protein L1	2.54	10.12	12.84
SERP0180	rplJ	fig|176279.9.peg.171	J	50S ribosomal protein L10	−1.40	2.38	2.07
SERP0186	rpsL	fig|176279.9.peg.177	J	30S ribosomal protein S12	1000.00	0.00	1000.00
SERP0187	rpsG	fig|176279.9.peg.178	J	30S ribosomal protein S7	11.20	11.48	13.52
SERP0188	fusA	fig|176279.9.peg.179	J	Elongation factor G	3.29	6.15	8.75
SERP0189	tuf	fig|176279.9.peg.180	J	Elongation factor Tu	−6.12	−2.74	−1.25
SERP0804	rpsP	fig|176279.9.peg.782	J	30S ribosomal protein S16	11.89	13.39	21.72
SERP0823	rpsB	fig|176279.9.peg.802	J	30S ribosomal protein S2	3.97	5.29	6.00
SERP0840	rpsO	fig|176279.9.peg.819	J	30S ribosomal protein S15	1000.00	1000.00	1000.00
SERP1116	rpmG1	fig|176279.9.peg.1087	J	50S ribosomal protein L33	0.00	1000.00	1000.00
SERP1153	rpsT	fig|176279.9.peg.1122	J	30S ribosomal protein S20	1.56	−1.17	3.74
SERP1211	rplU	fig|176279.9.peg.1180	J	50S ribosomal protein L21	10.30	16.59	19.34
SERP1242	rplT	fig|176279.9.peg.1209	J	50S ribosomal protein L20	1000.00	1000.00	1000.00
SERP1244	infC	fig|176279.9.peg.1211	J	Translation initiation factor IF-3	0.00	1000.00	1000.00
SERP1284	rpsD	fig|176279.9.peg.1252	J	30S ribosomal protein S4	−1000.00	33.17	295.23
SERP1728	rho	fig|176279.9.peg.1687	K	Transcription termination factor	0.00	0.00	1000.00
SERP1798	rpsI	fig|176279.9.peg.1755	J	30S ribosomal protein S9	5.07	12.83	17.64
SERP1799	rplM	fig|176279.9.peg.1756	J	50S ribosomal protein L13	3.05	6.68	5.74
SERP1804	rplQ	fig|176279.9.peg.1761	J	50S ribosomal protein L17	8.32	8.99	5.65
SERP1806	rpsK	fig|176279.9.peg.1763	J	30S ribosomal protein S11	49.19	41.70	202.55
SERP1807	rpsM	fig|176279.9.peg.1764	J	30S ribosomal protein S13	1000.00	1000.00	1000.00
SERP1809	infA	fig|176279.9.peg.1766	J	Translation initiation factor IF-1	2.91	3.65	4.95
SERP1812	rplO	fig|176279.9.peg.1769	J	50S ribosomal protein L15	9.94	14.43	15.32
SERP1813	rpmD	fig|176279.9.peg.1770	J	50S ribosomal protein L30	2.61	4.52	3.31
SERP1814	rpsE	fig|176279.9.peg.1771	J	30S ribosomal protein S5	−3.48	6.69	11.59
SERP1815	rplR	fig|176279.9.peg.1772	J	50S ribosomal protein L18	7.10	12.85	8.90
SERP1816	rplF	fig|176279.9.peg.1773	J	50S ribosomal protein L6	26.76	36.00	49.69
SERP1817	rpsH	fig|176279.9.peg.1774	J	30S ribosomal protein S8	3.32	6.06	11.55
SERP1819	rplE	fig|176279.9.peg.1776	J	50S ribosomal protein L5	2.03	9.30	11.50
SERP1820	rplX	fig|176279.9.peg.1777	J	50S ribosomal protein L24	2.71	2.98	2.39
SERP1821	rplN	fig|176279.9.peg.1778	J	50S ribosomal protein L14	−1.70	−1000.00	33.34
SERP1823	rpmC	fig|176279.9.peg.1780	J	50S ribosomal protein L29	4.32	6.88	9.17
SERP1824	rplP	fig|176279.9.peg.1781	J	50S ribosomal protein L16	−1000.00	−1000.00	72.77
SERP1825	rpsC	fig|176279.9.peg.1782	J	30S ribosomal protein S3	0.00	1000.00	1000.00
SERP1826	rplV	fig|176279.9.peg.1783	J	50S ribosomal protein L22	8.05	21.03	26.78
SERP1827	rpsS	fig|176279.9.peg.1784	J	30S ribosomal protein S19	19.81	14.13	13.71
SERP1828	rplB	fig|176279.9.peg.1785	J	50S ribosomal protein L2	0.00	1000.00	1000.00
SERP1830	rplD	fig|176279.9.peg.1787	J	50S ribosomal protein L4	0.00	1000.00	1000.00
SERP1831	rplC	fig|176279.9.peg.1788	J	50S ribosomal protein L3	6.47	40.98	40.65
SERP1832	rpsJ	fig|176279.9.peg.1789	J	30S ribosomal protein S10	2.53	4.45	15.72

**Table 10 cells-11-03488-t010:** Differentially expressed proteins associated with essential proteins.

					Fold Ratio		
Locus Tag	Gene	PATRIC ID	COG	Description	T1	T2	T3
SERP1040	SERP1040	fig|176279.9.peg.1014	-	Uncharacterized protein	−2.76	−2.77	1.38
SERP1041	hup	fig|176279.9.peg.1015	L	DNA-binding protein HU	−8.48	−1.26	−1.71
SERP1057	scpB	fig|176279.9.peg.1030	D	Segregation and condensation protein B	0.00	0.00	1000.00
SERP1076	SERP1076	fig|176279.9.peg.1047	C	Dihydrolipoamide acetyltransferase component of pyruvate dehydrogenase complex	0.00	0.00	1000.00
SERP1078	SERP1078	fig|176279.9.peg.1049	C	2-oxoisovalerate dehydrogenase subunit alpha	−2.59	−1.87	−1.89
SERP1082	ispA	fig|176279.9.peg.1053	H	Geranyltranstransferase	0.00	0.00	1000.00
SERP1090	argB	fig|176279.9.peg.1061	F	Acetylglutamate kinase	−1000.00	−1000.00	−1000.00
SERP1127	sigA	fig|176279.9.peg.1097	K	RNA polymerase sigma factor	3.91	4.99	27.12
SERP0120	tmk	fig|176279.9.peg.112	F	Thymidylate kinase	−1000.00	−1000.00	−1000.00
SERP1151	SERP1151	fig|176279.9.peg.1120	H	Heme chaperone	−1000.00	−1000.00	−1000.00
SERP1153	rpsT	fig|176279.9.peg.1122	J	30S ribosomal protein S20	1.56	−1.17	3.74
SERP0122	SERP0122	fig|176279.9.peg.114	L	DNA polymerase III subunit delta	−1000.00	−1000.00	−1000.00
SERP1193	hisS	fig|176279.9.peg.1162	J	Histidine--tRNA ligase	−1000.00	−1000.00	−1000.00
SERP1209	rpmA	fig|176279.9.peg.1178	J	50S ribosomal protein L27	5.89	4.72	1.09
SERP1211	rplU	fig|176279.9.peg.1180	J	50S ribosomal protein L21	10.30	16.59	19.34
SERP0128	metG	fig|176279.9.peg.120	J	Methionine--tRNA ligase	0.00	1000.00	1000.00
SERP1242	rplT	fig|176279.9.peg.1209	J	50S ribosomal protein L20	1000.00	1000.00	1000.00
SERP1246	thrS	fig|176279.9.peg.1213	J	Threonine--tRNA ligase	0.00	0.00	1000.00
SERP1252	coaE	fig|176279.9.peg.1220	F	Dephospho-CoA kinase	−2.36	−1.67	1.83
SERP1284	rpsD	fig|176279.9.peg.1252	J	30S ribosomal protein S4	−1000.00	33.17	295.23
SERP1297	SERP1297	fig|176279.9.peg.1265	E	Chorismate mutase/phospho-2-dehydro-3-deoxyheptonate aldolase	−1000.00	2.84	12.89
SERP1352	metK	fig|176279.9.peg.1319	H	S-adenosylmethionine synthase	0.00	1000.00	1000.00
SERP1367	cpfC	fig|176279.9.peg.1335	H	Coproporphyrin III ferrochelatase	−1000.00	−1000.00	1.44
SERP0149	hpt	fig|176279.9.peg.140	F	Hypoxanthine-guanine phosphoribosyltransferase	−4.04	1.27	1.97
SERP1437	gatB	fig|176279.9.peg.1405	J	Aspartyl/glutamyl-tRNA(Asn/Gln) amidotransferase subunit B	1.16	2.18	4.53
SERP1438	gatA	fig|176279.9.peg.1406	J	Glutamyl-tRNA(Gln) amidotransferase subunit A	3.56	4.41	14.25
SERP1439	gatC	fig|176279.9.peg.1407	J	Aspartyl/glutamyl-tRNA(Asn/Gln) amidotransferase subunit C	−9.71	−3.94	1.82
SERP1449	nadE	fig|176279.9.peg.1417	F	NH(3)-dependent NAD(+) synthetase	−1000.00	5.53	52.97
SERP1450	SERP1450	fig|176279.9.peg.1418	H	Nicotinate phosphoribosyltransferase	0.00	0.00	1000.00
SERP0168	gltX	fig|176279.9.peg.159	J	Glutamate--tRNA ligase	−1000.00	−1000.00	4.77
SERP1727	rpmE2	fig|176279.9.peg.1686	J	50S ribosomal protein L31 type B	1.19	−1.44	1.52
SERP0179	rplA	fig|176279.9.peg.170	J	50S ribosomal protein L1	2.54	10.12	12.84
SERP1743	deoB	fig|176279.9.peg.1702	G	Phosphopentomutase	−19.11	−4.55	−2.66
SERP0180	rplJ	fig|176279.9.peg.171	J	50S ribosomal protein L10	−1.40	2.38	2.07
SERP1760	glmS	fig|176279.9.peg.1717	M	Glutamine--fructose-6-phosphate aminotransferase	0.00	0.00	1000.00
SERP0181	rplL	fig|176279.9.peg.172	J	50S ribosomal protein L7/L12	1.81	2.02	1.16
SERP1777	SERP1777	fig|176279.9.peg.1733	P	Iron compound ABC transporter, iron compound-binding protein	22.55	19.37	10.98
SERP0183	rpoB	fig|176279.9.peg.174	K	DNA-directed RNA polymerase subunit beta	16.65	9.65	267.97
SERP0184	rpoC	fig|176279.9.peg.175	K	DNA-directed RNA polymerase subunit beta	0.00	1000.00	1000.00
SERP1798	rpsI	fig|176279.9.peg.1755	J	30S ribosomal protein S9	5.07	12.83	17.64
SERP1799	rplM	fig|176279.9.peg.1756	J	50S ribosomal protein L13	3.05	6.68	5.74
SERP1804	rplQ	fig|176279.9.peg.1761	J	50S ribosomal protein L17	8.32	8.99	5.65
SERP1805	rpoA	fig|176279.9.peg.1762	K	DNA-directed RNA polymerase subunit alpha	1.00	2.04	6.58
SERP1806	rpsK	fig|176279.9.peg.1763	J	30S ribosomal protein S11	49.19	41.70	202.55
SERP1807	rpsM	fig|176279.9.peg.1764	J	30S ribosomal protein S13	1000.00	1000.00	1000.00
SERP1810	adk	fig|176279.9.peg.1767	F	Adenylate kinase	2.04	2.50	2.89
SERP1812	rplO	fig|176279.9.peg.1769	J	50S ribosomal protein L15	9.94	14.43	15.32
SERP0186	rpsL	fig|176279.9.peg.177	J	30S ribosomal protein S12	1000.00	0.00	1000.00
SERP1813	rpmD	fig|176279.9.peg.1770	J	50S ribosomal protein L30	2.61	4.52	3.31
SERP1814	rpsE	fig|176279.9.peg.1771	J	30S ribosomal protein S5	−3.48	6.69	11.59
SERP1815	rplR	fig|176279.9.peg.1772	J	50S ribosomal protein L18	7.10	12.85	8.90
SERP1816	rplF	fig|176279.9.peg.1773	J	50S ribosomal protein L6	26.76	36.00	49.69
SERP1817	rpsH	fig|176279.9.peg.1774	J	30S ribosomal protein S8	3.32	6.06	11.55
SERP1819	rplE	fig|176279.9.peg.1776	J	50S ribosomal protein L5	2.03	9.30	11.50
SERP1820	rplX	fig|176279.9.peg.1777	J	50S ribosomal protein L24	2.71	2.98	2.39
SERP1821	rplN	fig|176279.9.peg.1778	J	50S ribosomal protein L14	−1.70	−1000.00	33.34
SERP0187	rpsG	fig|176279.9.peg.178	J	30S ribosomal protein S7	11.20	11.48	13.52
SERP1823	rpmC	fig|176279.9.peg.1780	J	50S ribosomal protein L29	4.32	6.88	9.17
SERP1824	rplP	fig|176279.9.peg.1781	J	50S ribosomal protein L16	−1000.00	−1000.00	72.77
SERP1825	rpsC	fig|176279.9.peg.1782	J	30S ribosomal protein S3	0.00	1000.00	1000.00
SERP1826	rplV	fig|176279.9.peg.1783	J	50S ribosomal protein L22	8.05	21.03	26.78
SERP1827	rpsS	fig|176279.9.peg.1784	J	30S ribosomal protein S19	19.81	14.13	13.71
SERP1828	rplB	fig|176279.9.peg.1785	J	50S ribosomal protein L2	0.00	1000.00	1000.00
SERP1830	rplD	fig|176279.9.peg.1787	J	50S ribosomal protein L4	0.00	1000.00	1000.00
SERP1831	rplC	fig|176279.9.peg.1788	J	50S ribosomal protein L3	6.47	40.98	40.65
SERP1832	rpsJ	fig|176279.9.peg.1789	J	30S ribosomal protein S10	2.53	4.45	15.72
SERP2056	gtaB	fig|176279.9.peg.2003	M	UTP--glucose-1-phosphate uridylyltransferase	−2.61	1.31	−1.00
SERP2122	SERP2122	fig|176279.9.peg.2065	I	Hydroxymethylglutaryl-CoA synthase	−1000.00	−1000.00	−1000.00
SERP2191	cysJ	fig|176279.9.peg.2133	C	Sulfite reductase (NADPH) flavoprotein alpha-component	0.00	0.00	1000.00
SERP2534	walR	fig|176279.9.peg.2453	K	Transcriptional regulatory protein	−1000.00	17.76	−1000.00
SERP2538	rplI	fig|176279.9.peg.2456	J	50S ribosomal protein L9	1.09	−1.36	−1.79
SERP2545	serS	fig|176279.9.peg.2463	J	Serine--tRNA ligase	−3.87	−2.06	−1.08
SERP2552	dnaN	fig|176279.9.peg.2469	L	Beta sliding clamp	−1000.00	−10.91	−1.07
SERP0262	argS	fig|176279.9.peg.250	J	Arginine--tRNA ligase	−1000.00	4.25	55.70
SERP0405	murB	fig|176279.9.peg.385	M	UDP-N-acetylenolpyruvoylglucosamine reductase	0.00	0.00	1000.00
SERP0044	rpsF	fig|176279.9.peg.43	J	30S ribosomal protein S6	1.81	1.09	−1.09
SERP0046	rpsR	fig|176279.9.peg.45	J	30S ribosomal protein S18	0.00	0.00	1000.00
SERP0518	dltA	fig|176279.9.peg.496	Q	D-alanine--D-alanyl carrier protein ligase	0.00	1000.00	0.00
SERP0520	dltC	fig|176279.9.peg.498	IQ	D-alanyl carrier protein	−1000.00	−106.80	−1.94
SERP0568	fabF	fig|176279.9.peg.549	I	3-oxoacyl-[acyl-carrier-protein] synthase 2	−16.63	−1.86	1.00
SERP0575	trpS	fig|176279.9.peg.556	J	Tryptophan--tRNA ligase	0.00	0.00	1000.00
SERP0629	SERP0629	fig|176279.9.peg.610	HQ	Isochorismate synthase	−1000.00	−1000.00	−1000.00
SERP0632	menB	fig|176279.9.peg.613	H	1,4-dihydroxy-2-naphthoyl-CoA synthase	0.00	1000.00	1000.00
SERP0721	pheS	fig|176279.9.peg.700	J	Phenylalanine--tRNA ligase alpha subunit	−1000.00	−1000.00	−13.26
SERP0758	ileS	fig|176279.9.peg.737	J	Isoleucine--tRNA ligase	0.00	0.00	1000.00
SERP0776	gmk	fig|176279.9.peg.754	F	Guanylate kinase	0.00	0.00	1000.00
SERP0796	fabD	fig|176279.9.peg.774	I	Malonyl CoA-acyl carrier protein transacylase	−1000.00	−1.14	3.27
SERP0804	rpsP	fig|176279.9.peg.782	J	30S ribosomal protein S16	11.89	13.39	21.72
SERP0823	rpsB	fig|176279.9.peg.802	J	30S ribosomal protein S2	3.97	5.29	6.00
SERP0833	nusA	fig|176279.9.peg.812	K	Transcription termination/antitermination protein NusA	−4.89	−1.25	4.23
SERP0840	rpsO	fig|176279.9.peg.819	J	30S ribosomal protein S15	1000.00	1000.00	1000.00
SERP0912	tkt	fig|176279.9.peg.890	G	Transketolase	−1000.00	−2.32	3.24
SERP1024	asnS	fig|176279.9.peg.998	J	Asparagine--tRNA ligase	−1000.00	−1000.00	−4.18

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
