# Peer review of "Concentration-Dependent Global Quantitative Proteome Response of Staphylococcus epidermidis RP62A Biofilms to Subinhibitory Tigecycline"

_cells, 2022, doi:10.3390/cells11213488_

Round 1
Reviewer 1 Report
In the manuscript "Concentration-dependent global quantitative proteome response of Staphylococcus epidermidis RP62A biofilms to subinhibitory tigecycline" K Sung, M Park, J Chon, O Kweon, SA Khan, A Shen and A Paredes investigated the effect of subinhibitory concentrations of tigecycline on S. epidermidis (strain RP62A) biofilms by a quantitative global proteomic technique. This study is interesting because highlights that global protein expression profiling of biofilm cells to antibiotic pressure may improve our understanding of the mechanisms of antibiotic resistance in biofilms.
Here some of the major and minor comments:
Line 5-13: The authors’ list and affiliations must be amended following the journal indication.
The authors did not show any MIC data, although they indicated in the material and methods section and the results in the text. I think it is necessary to show a graph/table about this since the entire MS is based on different dilution of MIC. Also the authors should indicate how they measured and show related graph indicating the number of live and death cells.
Line 144-145: “FESEM with low magnification clearly showed that 1/8 and 1/4 MIC TC treatment promoted biofilm formation.” not sure whether this affirmation is correct. All the affirmation based on CSLM and FESEM are purely qualitative and not quantitative. At least for CSLM the authors may reinforce the evidence by analyzing the fluorescence intensity for example.
Line 150..: regarding the MV, the authors indicated MV only by morphological FESEM, what about the use of some specific marker?
The section “Results and discussion” is 15 page long, starting from line 140 to 338, honestly I found this too long and complicate to follow hence understand. I would suggest to separate results from discussion, this may help understanding more comprehensibly all the evidences and the respective drawn interpretations.
In particular, it is very hard to follow the part regarding the proteome, is too long and with many repetitions that make the reading very hard especially for a not expert reader in this branch. I suggest to rewrite entirely separating result from discussion and separating the results in different more readily paragraphs. Moreover, I would suggest to report the data as heatmap, this may result in a clearer and immediate way to understand the figure and data.
Moreover, the statistics is completely missing.
Author Response
Reviewer 1
In the manuscript "Concentration-dependent global quantitative proteome response of Staphylococcus epidermidis RP62A biofilms to subinhibitory tigecycline" K Sung, M Park, J Chon, O Kweon, SA Khan, A Shen and A Paredes investigated the effect of subinhibitory concentrations of tigecycline on S. epidermidis (strain RP62A) biofilms by a quantitative global proteomic technique. This study is interesting because highlights that global protein expression profiling of biofilm cells to antibiotic pressure may improve our understanding of the mechanisms of antibiotic resistance in biofilms.
Authors highly appreciate the reviewer’s thorough reading of the manuscript and critical remarks that helped us to improve the manuscript. We have seriously considered and addressed your valuable comments by point-to-point responses in the following context. The reviewer’s concern was reflected in the revised manuscript. We hope our revision has improved the research article to a level of your satisfaction.
Here some of the major and minor comments:
Comment: Line 5-13: The authors’ list and affiliations must be amended following the journal indication.
Response: Thank you for your valuable comments/suggestions. The authors corrected the authors’ list and affiliations.
Comment: The authors did not show any MIC data, although they indicated in the material and methods section and the results in the text. I think it is necessary to show a graph/table about this since the entire MS is based on different dilution of MIC. Also the authors should indicate how they measured and show related graph indicating the number of live and death cells.
Response: Thank you for your valuable comments/suggestions. The authors added MIC graph (Fig. 1) to the manuscript as the reviewer suggested.
Graph of the live cell count (colony forming unit/ml, Figure 3) was also added to the manuscript.
Comment: Line 144-145: “FESEM with low magnification clearly showed that 1/8 and 1/4 MIC TC treatment promoted biofilm formation.” not sure whether this affirmation is correct. All the affirmation based on CSLM and FESEM are purely qualitative and not quantitative. At least for CSLM the authors may reinforce the evidence by analyzing the fluorescence intensity for example.
Response: Thank you for your valuable comments/suggestions. Graph of the live cell count (colony forming unit/ml, Figure 3) showed quantitatively the promotion of biofilm production in 1/8 and 1/4 MIC TC-treated samples compared to the 1/2 MIC TC-treated sample. Therefore, we didn’t measure the additional fluorescence intensity.
Comment: Line 150: regarding the MV, the authors indicated MV only by morphological FESEM, what about the use of some specific marker?
Response: Thank you for your valuable comments/suggestions. Lee et al. (Gram-positive bacteria produce membrane vesicles: Proteomics-based characterization of Staphylococcus aureus-derived membrane vesicles, Proteomics, 2009, 9, 5425–5436) purified membrane vesicle (MV) from Staphylococcus aureus strain and identified a total of 90 proteins. Although the bacterial species were different, and the proteins were identified from MV (S. aureus) and whole cell proteins (S. epidermidis RP62A), almost half (46.7%, 42/90) the same proteins were identified from S. epidermidis RP62A proteome. This indicates that S. epidermidis RP62A secreted MV.
Furthermore, many research groups confirmed the presence of MV using SEM only. Please find the research articles for your reference.
- Membrane Vesicles Are the Dominant Structural Components of Ceftazidime-Induced Biofilm Formation in an Oxacillin-Sensitive MRSA. Front Microbiol. 2019;10:571
- Extracellular vesicles derived from Staphylococcus aureus induce atopic dermatitis-like skin inflammation. Allergy. 2011;66(3):351-9
- Membrane vesicles, nanopods and/or nanotubes produced by hyperthermophilic archaea of the genus Thermococcus. Biochem Soc Trans. 2013;41(1):436-42
- Detection and Physicochemical Characterization of Membrane Vesicles (MVs) of Lactobacillus reuteri DSM 17938, Front Microbiol. 2017;8:1040
- Isolation and Characterization of Outer Membrane Vesicles of Pectobacterium brasiliense 1692. Microorganisms. 2021;9(9):1918
- Modeled microgravity alters lipopolysaccharide and outer membrane vesicle production of the beneficial symbiont Vibrio fischeri. NPJ Microgravity. 2021;7(1):8
- Characterization of membrane vesicles released by Mycobacterium avium in response to environment mimicking the macrophage phagosome. Future Microbiol. 2019;14(4):293-313
- Abiotic stressors impact outer membrane vesicle composition in a beneficial rhizobacterium: Raman spectroscopy characterization. Sci Rep. 2020;10(1):21289
Comment: The section “Results and discussion” is 15 page long, starting from line 140 to 338, honestly I found this too long and complicate to follow hence understand. I would suggest to separate results from discussion, this may help understanding more comprehensibly all the evidences and the respective drawn interpretations.
Response: Thank you for your valuable comments/suggestions. Authors agree the reviewer’s suggestion and separated results from discussion.
Comment: In particular, it is very hard to follow the part regarding the proteome, is too long and with many repetitions that make the reading very hard especially for a not expert reader in this branch. I suggest to rewrite entirely separating result from discussion and separating the results in different more readily paragraphs. Moreover, I would suggest to report the data as heatmap, this may result in a clearer and immediate way to understand the figure and data.
Response: Thank you for your valuable comments/suggestions. Authors agree the reviewer’s suggestion and separated results from discussion. In addition, we added a heatmap figure (Figure 6) with row (based on COG functional groups) and column annotations (control vs. TC treatment) and column clustering (sample-based) which shows the correlation between protein expression profiles and TC concentrations treated. We have updated the method section related to the heatmap.
Fig. 6. Heatmap presenting differentially expressed proteins. The proteins in the heatmap analysis were clustered according to five COG functional groups; Cel (Cellular process and signaling), Inf (Information storage and processing), Met (Metabolism), No (No COG annotation), and Poo (Poorly characterized).
Comment: Moreover, the statistics is completely missing.
Response: Thank you for your valuable comments/suggestions. We agree that replication is key to reducing the variability of experimental results. We also believe that design principles, such as a time course or concentration-dependent experiments, are widely used to investigate the proteome with acceptable confidence. This proteomic study was not replicated but has adopted a concentration-dependent experiment design. To address the reviewer’s comments, we further analyzed the proteomic data in terms of expression pattern and functional enrichment and added a figure (Figure 12) and its description to the manuscript.
Fig. 12. Protein expression patterns and functional distribution of proteins showing TC-dependent expression changes.
We added the description: “Protein expression pattern analysis tool that is integrated with COG functional category analysis, was used to cluster proteins that show a similar temporal expression pattern. In total, 27 significant temporal protein expression profiles were clustered as shown in Fig. 12. When strain RP62A biofilms were treated with TC, about 100 proteins involved in information storage and processing (INF; J, K, and L), cellular processes and signaling (CEL; D, M, U, and O), and metabolism (MET; C, G, E, and P) were significantly upregulated (EPN22). Notably, translation, ribosomal structure and biogenesis (J) accounted for the largest portion (30 proteins, 40.5%) of the upregulated proteins. About 48 proteins included in EPN1 were consistently identified in all TC-treated biofilm samples including control, and the majority of them consisted of translation, ribosomal structure and biogenesis (J), posttranslational modification, protein turnover, chaperones (O), and carbohydrate transport and metabolism (G). As revealed in EPN2, proteins associated with INF (J and L) and MET (C, G, E, F, H, I, P, and Q) were substantially downregulated in all TC-treated biofilms (T1, T2, and T3). Protein expression patterns that were upregulated (EPN19) or downregulated (EPN9) only at the highest TC concentration were confirmed”.

Reviewer 2 Report
In this study Sung et al reports an extensive proteome study on the effects of sub MIC concentrations of tigecycline on Staphylococcus epidermidis biofilm. A larger number of changes in the levels of a variety of individual proteins are identified and compared to findings in the literature as well as superficially discussed in terms of function. However, overall this is almost exclusively a descriptive, one dimensional study, as is also evident from the fact that virtually no (mechanistic) conclusions are drawn. In addition, none of the protein level changes have been qualified by independent methods. Thus, further data allowing more mechanistic insight is required for publication. For instance, an obvious comparison would be parallel analogous data for bacteria in planktonic culture, if one wishes to identify biofilm related effects, as would more detailed exploration of key protein changes using independent methods.
In many places throughout the paper, the text is unclear, as exemplified:
Line 53: What is the definition of “Sublethal antibiotics”
Line 57-58: please rephrase “have influenced the expression of some virulence factors”
Line 69: What are “extremely sophisticated sensitivities”
Line 93: How is “visible cell growth determined spectrophotometrically” ?
Line 166: “A marked proteome was induced from the lowest TC concentration” What is a marked proteome?
Line 167: How are “PR62A biofilms (are) able to sense low concentrations of TC”?
Line 171: “. These results indicate that metabolism is critical for the resistance of strain RP62A biofilms to TC treatment”. Is metabolism not critical for most cellular functions?
Line 189: “indicating there were difficulties in interpreting the proteome data.” Is this not a problem for all the data?
Line 197-9: “These results indicate that glycolysis, the tricarboxylic acid (TCA) and pentose 197 phosphate pathways, pyruvate oxidation, purine and pyrimidine metabolism, and amino acid and cofactor biosynthesis 198 may play key roles assisting strain RP62A in defending against antibiotic pressure.” This seems an obvious observation to expand on?!
Line 213: “Sub-MIC TC concentrations modified 12 proteins involved in biofilm formation”. Modified indicates protein modification. Which were they?
Line 243: “genes resistant to several antibiotics” a gene cannot be resistant to antibiotics.
Line 252: “This result is inconsistent with previous findings that A. baumannii grown with subinhibitory TC remarkably suppressed β-lactam resistance 253 genes [44,45]” What is the conclusion from this?
Line 316: “RpsR, a ribosomal protein S18, was not detected in T1 and T2 biofilms”. Which alternative S18 protein is this? It is highly improbable (or interesting) if one of the 30S subunit proteins disappear in a living cell.
Author Response
Reviewer 2
In this study Sung et al reports an extensive proteome study on the effects of sub MIC concentrations of tigecycline on Staphylococcus epidermidis biofilm. A larger number of changes in the levels of a variety of individual proteins are identified and compared to findings in the literature as well as superficially discussed in terms of function. However, overall this is almost exclusively a descriptive, one dimensional study, as is also evident from the fact that virtually no (mechanistic) conclusions are drawn. In addition, none of the protein level changes have been qualified by independent methods. Thus, further data allowing more mechanistic insight is required for publication. For instance, an obvious comparison would be parallel analogous data for bacteria in planktonic culture, if one wishes to identify biofilm related effects, as would more detailed exploration of key protein changes using independent methods.
Authors highly appreciate the reviewer’s thorough reading of the manuscript and critical remarks that helped us to improve the manuscript. We have seriously considered and addressed your valuable comments by point-to-point responses in the following context. The reviewer’s concern was reflected in the revised manuscript. We hope our revision has improved the research article to a level of your satisfaction.
Comment: ……None of the protein level changes have been qualified by independent methods.
Response: Thank you for your valuable comments/suggestions. This study was an initial study to investigate mainly a top-down insight into tigecycline concentration-dependent phenotypic and proteomic alterations. We have a plan to conduct a series of enzyme-centric bottom-up experiments on proteins showing tigecycline-dependent protein expression changes. We hope these endeavors allow us to bridge the gaps between the proteomic data and phenotypic changes.
In many places throughout the paper, the text is unclear, as exemplified:
Comment: Line 53: What is the definition of “Sublethal antibiotics”
Response: Thank you for your valuable comments/suggestions. “Sublethal antibiotics” was corrected as “Subinhibitory antibiotics”.
Comment: Line 57-58: please rephrase “have influenced the expression of some virulence factors”
Response: Thank you for your valuable comments/suggestions. Line 57-58 was corrected as “reduced the expression of Staphylococcus aureus virulence factors”.
Comment: Line 69: What are “extremely sophisticated sensitivities”
Response: Thank you for your valuable comments/suggestions. The phrase means that the Q-Exactive HF-X Orbitrap mass spectrometer is equipped with a high-capacity transmission tube that enables an internal mass accuracy with less than 1 ppm.
Comment: Line 93: How is “visible cell growth determined spectrophotometrically” ?
Response: Thank you for your valuable comments/suggestions. Bacterial cells were grown with various concentrations of antibiotics in a 96-well plate and measured their growth by measuring their optical density at 600 nm in a spectrophotometer (a Synergy 2 Multi-Mode Microplate Reader). After 24 hour growth, authors can check the cell growth with a kinetic growth curve measured by the spectrophotometer. In addition, we can confirm the cell growth with a visually clear broth (No cell growth). To avoid confusion, authors corrected as “The lowest concentration of antibiotic with no visible cell growth was defined as the MIC”.
Comment: Line 166: “A marked proteome was induced from the lowest TC concentration” What is a marked proteome?
Response: Thank you for your valuable comments/suggestions. Following treatment with the lowest tigecycline (TC) concentration (0.031 µg/ml) on S. epidermidis RP62A biofilm, 413 proteins (165 upregulated and 248 downregulated) were differentially expressed. Using “a marked proteome’, authors wanted to emphasize that a significant number of proteins were expressed even when the lowest concentration of tigecycline (TC) was treated. Authors corrected as “a significant number of proteins were induced at the lowest TC concentration (0.031 µg/ml)”.
Comment: Line 167: How are “PR62A biofilms (are) able to sense low concentrations of TC”?
Response: Thank you for your valuable comments/suggestions. Following treatment with low concentrations of tigecycline (TC) on S. epidermidis RP62A biofilm, a significant number of proteins (T1: 413, T2: 429, T3: 518) were differentially expressed. These results indicate strain RP62A biofilms can sense and respond to low concentrations of tigecycline (TC). Authors corrected as “PR62A biofilms are able to respond to low concentrations of TC”.
Comment: Line 171: “. These results indicate that metabolism is critical for the resistance of strain RP62A biofilms to TC treatment”. Is metabolism not critical for most cellular functions?
Response: Thank you for your valuable comments/suggestions. Authors agree that metabolism is critical for most cellular functions. Fifty-two proteins were detected with the lowest fold ratios (fold ratio -1,000) in all sublethal tigecycline (TC)-treated biofilms and they belonged to the metabolism pathway accounting for 90.8% (Cellular Processes pathways: 0%, Environmental Information Processing pathways: 3.4%, Genetic Information Processing pathways: 5.0%, Human Diseases pathways: 0.8%). Based on the observation, authors stated that metabolism is critical for the resistance of strain RP62A biofilms to TC treatment.
Comment: Line 189: “indicating there were difficulties in interpreting the proteome data.” Is this not a problem for all the data?
Response: Thank you for your valuable comments/suggestions. Cluster of Orthologous Groups (COG) is a database consisting of functional annotation of gene classification. Function unknown (S) category of COG in S. epidermidis RP62A accounted for 24.8% (T1), 20.2% (T2), and 22.0% (T3) in upregulated proteins, and 13.2% (T1), 13.5% (T2), and 16.0% (T3) in downregulated proteins, respectively. Authors agree that it is not easy to interpret the whole proteome data by COG only. To better understand the proteome data, authors performed both COG and KEGG pathway analyses. In addition, we further analyzed the proteomic data in terms of expression pattern and functional enrichment and added a figure (Figure 12) and its description to the manuscript.
Fig. 12. Protein expression patterns and functional distribution of proteins showing TC-dependent expression changes.
Comment: Line 197-9: “These results indicate that glycolysis, the tricarboxylic acid (TCA) and pentose 197 phosphate pathways, pyruvate oxidation, purine and pyrimidine metabolism, and amino acid and cofactor biosynthesis 198 may play key roles assisting strain RP62A in defending against antibiotic pressure.” This seems an obvious observation to expand on?!
Response: Thank you for your valuable comments/suggestions. Authors corrected as “Within the two categories, a significant numbers of the differentially expressed proteins associated with glycolysis (39 proteins), the tricarboxylic acid (TCA) and pentose phosphate pathways (51 proteins), pyruvate oxidation (51 proteins), purine and pyrimidine metabolism (56 proteins), and amino acid and cofactor biosynthesis (219 proteins) were identified after TC treatment, indicating that these protein groups may play key roles assisting strain RP62A in defending against antibiotic pressure”.
Comment: Line 213: “Sub-MIC TC concentrations modified 12 proteins involved in biofilm formation”. Modified indicates protein modification. Which were they?
Response: Thank you for your valuable comments/suggestions. Authors intended to state that sub-MIC TC concentrations changed expression of 12 proteins involved in biofilm formation. To avoid confusion, authors corrected as “Sub-MIC TC concentrations changed expression of 12 proteins involved in biofilm formation”.
Comment: Line 243: “genes resistant to several antibiotics” a gene cannot be resistant to antibiotics.
Response: Thank you for your valuable comments/suggestions. Authors agree the reviewer’s point and corrected as “several antibiotic resistant genes”.
Comment: Line 252: “This result is inconsistent with previous findings that A. baumannii grown with subinhibitory TC remarkably suppressed β-lactam resistance 253 genes [44,45]” What is the conclusion from this?
Response: Thank you for your valuable comments/suggestions. Authors of references 44 (Li L, Hassan KA, Tetu SG, Naidu V, Pokhrel A, Cain AK, Paulsen IT. The Transcriptomic Signature of Tigecycline in Acinetobacter baumannii. Front Microbiol. 2020;11:565438) and 45 (Hua X, Chen Q, Li X, Yu Y. Global transcriptional response of Acinetobacter baumannii to a subinhibitory concentration of tigecycline. Int J Antimicrob Agents. 2014;44(4):337-44) reported that β-lactam resistance genes, such as OXA-23 and AmpC, were downregulated in the presence of tigecycline (TC). They employed transcriptomic technique (RNA sequencing) but we used the proteomic technique. Furthermore, they used a Gram-negative bacteria (Acinetobacter baumannii) but we used a Gram-positive bacteria (S. epidermidis RP62A). Therefore, it is very hard to conclude. We may conclude that TC may influence expression of β-lactam resistance genes/proteins.
Comment: Line 316: “RpsR, a ribosomal protein S18, was not detected in T1 and T2 biofilms”. Which alternative S18 protein is this? It is highly improbable (or interesting) if one of the 30S subunit proteins disappear in a living cell.
Response: Thank you for your valuable comments/suggestions. In 50S ribosomal proteins, twenty-four were identified. Of those, two proteins (RplB, RplD) were not detected in T1 biofilm but the remaining proteins were differentially expressed in all biofilms. In 30S ribosomal proteins, sixteen were identified. Of those, two proteins (RpsC, RpsL) were not detected in T1 and T2 biofilms, respectively, and RpsR was not detected in T1 and T2 biofilms.

Round 2
Reviewer 1 Report
Authors provided appropriate correction/justification to my concerns and extensively revise the text. Nevertheless, Still not understand the MIC experiment, The Fig1 does not report the MIC but the growth curve over 24h culture with different concentration of TC. Moreover, the Fig legend miss to indicate what some symbols represent.
Author Response
Reviewer 1
Authors provided appropriate correction/justification to my concerns and extensively revise the text. Nevertheless, Still not understand the MIC experiment, The Fig1 does not report the MIC but the growth curve over 24h culture with different concentration of TC. Moreover, the Fig legend miss to indicate what some symbols represent.
Response: Thank you for your valuable comments/suggestions. Figure 1 shows the growth curve of Staphylococcus epidermidis RP62A in serially-diluted tigecycline and indicates the bacteria was not grown at 0.25 µg/ml tigecycline. That means the Minimum Inhibitory Concentration (MIC) of tigecycline is 0.25 µg/ml. Please find reading results of a plate reader (Row A: Mueller-Hinton broth control without TC, Rows B, C, D: Mueller-Hinton broth with TC (three replicates), Column 1: Mueller-Hinton broth without S. epidermidis RP62A (Blank control), columns 2-12: 8, 4, 2, 1, 0.5, 0.25, 0.125, 0.0625, 0.031, 0.016, 0.008 µg/ml TC). The plate reader reading also indicates that the MIC of TC is 0.25 µg/ml (Column 7). Authors added missing legend symbols of Figure 1.

Reviewer 2 Report
The paper has been improved
Author Response
Authors appreciate the reviewer's comments that the manuscript has been improved.